# The 133-kDa N-terminal domain enables myosin 15 to maintain mechanotransducing stereocilia and is essential for hearing

Qing Fang[1], Artur A Indzhykulian[2†], Mirna Mustapha[1‡], Gavin P Riordan[3], David F Dolan[4], Thomas B Friedman[3], Inna A Belyantseva[3], Gregory I Frolenkov[2], Sally A Camper[1], Jonathan E Bird[3*]

[1]Department of Human Genetics, University of Michigan, Ann Arbor, United States; [2]Department of Physiology, University of Kentucky, Lexington, United States; [3]Laboratory of Molecular Genetics, National Institute on Deafness and Other Communication Disorders, National Institutes of Health, Bethesda, United States; [4]Department of Otolaryngology, University of Michigan Medical School, Ann Arbor, United States

*For correspondence: jonathan. bird@nih.gov

Present address: †Department of Neurobiology, Harvard Medical School, Boston, United States; ‡Department of Otolaryngology—Head and Neck Surgery, Stanford University, Stanford, United States

Competing interests: The authors declare that no competing interests exist.

**Abstract** The precise assembly of inner ear hair cell stereocilia into rows of increasing height is critical for mechanotransduction and the sense of hearing. Yet, how the lengths of actin-based stereocilia are regulated remains poorly understood. Mutations of the molecular motor myosin 15 stunt stereocilia growth and cause deafness. We found that hair cells express two isoforms of myosin 15 that differ by inclusion of an 133-kDa N-terminal domain, and that these isoforms can selectively traffic to different stereocilia rows. Using an isoform-specific knockout mouse, we show that hair cells expressing only the small isoform remarkably develop normal stereocilia bundles. However, a critical subset of stereocilia with active mechanotransducer channels subsequently retracts. The larger isoform with the 133-kDa N-terminal domain traffics to these specialized stereocilia and prevents disassembly of their actin core. Our results show that myosin 15 isoforms can navigate between functionally distinct classes of stereocilia, and are independently required to assemble and then maintain the intricate hair bundle architecture.

## Introduction

The inner ear detects sound using mechanosensitive hair bundles that project from the apical surface of cochlear hair cells (reviewed in *Schwander et al., 2010*). Each hair bundle is composed of actin-based stereocilia that are arranged into rows of increasing height to create a staircase-like architecture; a feature evolutionarily conserved in vertebrates (*Manley, 2000*). Extracellular tip-link filaments connect the tip of each stereocilium to the lateral shaft of its taller neighbor (*Pickles et al., 1984*). Tensioning of these links during hair bundle deflection initiates mechanoelectrical transduction (MET) by gating mechanotransducer ion channels that are located at the tips of shorter rows of stereocilia (*Beurg et al., 2009*). Formation of the mature staircase architecture involves a complex program of differential elongation and thickening of the individual stereocilia actin cores (*Tilney et al., 1988*). How this process is developmentally specified with sub-nanometer tolerances and subsequently maintained throughout adult life is poorly understood.

Unconventional myosin 15 (encoded by *Myo15*) is an actin-based molecular motor and a key regulator of hair bundle development. In mice with mutations of the myosin 15 motor or tail domain, *shaker 2* (*Myo15sh2/sh2*) or *shaker 2-J* (*Myo15sh2-J/sh2-J*), respectively, hair bundles are short and this results in profound hearing loss and vestibular dysfunction (*Probst et al., 1998*; *Anderson et al., 2000*). Mutations in the human ortholog *MYO15A* similarly cause non-syndromic autosomal recessive deafness,

**eLife digest** Sound is detected by the cochlea, a coiled structure encapsulated within the inner ear of humans and other mammals. Inside this organ, intricate arrays of sensory hair cells are stimulated by sound to generate neural signals that are transmitted to the brain. The 'hairs' that give hair cells their name are actually structures called stereocilia that act like antennas to detect sound waves. Damage to these delicate mechanical sensors, through genetic mutations or loud noise, are a significant cause of hearing loss in humans.

A protein called myosin 15 is a molecular motor needed for stereocilia to develop and grow to their normal height. Mutations of this protein cause hereditary deafness in humans. Hair cells produce two versions of myosin 15, which are identical except for one version having an extra region called the N-terminal extension.

Using genetic engineering, Fang et al. created mutant mice that only produce the smaller version of myosin 15 that lack the N-terminal extension. These mutant mice helped reveal that when hair cells are young, they mostly produce the smaller version of myosin 15, and this is sufficient for stereocilia to grow normally. Once hair cells mature however, they switch to producing the larger version of myosin 15 that contains the N-terminal extension. In the mutant mice that lacked the larger version of myosin 15, stereocilia ultimately deteriorate, leaving the hair cells unable to detect sound.

Myosin 15 was previously known to help stereocilia grow, but Fang et al. now show that this protein is also required to maintain stereocilia throughout life. The next challenge is to understand how the N-terminal extension enables myosin 15 to preserve the structure of adult stereocilia, and to investigate whether this activity might be stimulated to prevent hearing loss.

DFNB3 (*Friedman et al., 1995*; *Wang et al., 1998*). Myosin 15 localizes to the tips of stereocilia (*Belyantseva et al., 2003*; *Rzadzinska et al., 2004*; *Belyantseva et al., 2005*), a site of barbed-end actin filament growth and turnover (*Schneider et al., 2002*; *Zhang et al., 2012*; *Drummond et al., 2015*; *Narayanan et al., 2015*). Myosin 15 is required for stereocilia elongation and traffics molecules to the stereocilia tips, including whirlin, a cytoskeletal scaffolding protein (*Mburu et al., 2003*; *Belyantseva et al., 2005*; *Delprat et al., 2005*), and epidermal growth factor receptor pathway substrate 8 (Eps8) which has actin binding, bundling and barbed-end capping activity (*Disanza et al., 2004*; *Manor et al., 2011*). The loss of either whirlin or Eps8 recapitulates the short hair bundle phenotype and deafness of $Myo15^{sh2/sh2}$ mice (*Mburu et al., 2003*; *Belyantseva et al., 2005*; *Manor et al., 2011*; *Zampini et al., 2011*), consistent with these proteins forming a complex with myosin 15 to promote stereocilia growth.

Alternative splicing creates two major protein isoforms from the 66 exon *Myo15* gene (*Liang et al., 1999*). Isoform 2 transcripts skip exon 2 and use a translation start codon in exon 3 to encode a 262 kDa protein including the motor ATPase domain and C-terminal MyTH4, SH3 and FERM moieties (*Figure 1A*). Isoform 1 transcripts include exon 2 that contains an alternate translation start codon and adds a 133-kDa N-terminal extension in frame with the motor domain and tail (*Figure 1A*). Both isoform transcripts are detected in inner ear cDNAs (*Belyantseva et al., 2003*) and are expressed by hair cells (*Liang et al., 1999*; *Anderson et al., 2000*; *Caberlotto et al., 2011*). Overexpression of isoform 2 can induce stereocilia elongation in $Myo15^{sh2/sh2}$ cochleae in vitro (*Belyantseva et al., 2005*), but the function of isoform 1 remains unknown. However, given that mutations in exon 2 are associated with DFNB3 deafness in humans, it strongly suggests that isoform 1 also has a critical role in the auditory system (*Nal et al., 2007*; *Cengiz et al., 2010*; *Bashir et al., 2012*; *Fattahi et al., 2012*).

In this study, we show that both isoforms of myosin 15 are expressed in auditory hair cells at different developmental stages, and that they traffic to distinct sub-cellular locations within the stereocilia hair bundle. To understand their individual functions, we engineered a mouse model carrying a nonsense mutation in exon 2 that ablates isoform 1, leaving isoform 2 intact. We found that hair bundles depend critically upon two phases of myosin 15 activity throughout their lifetime; isoform 2 orchestrates development of the staircase architecture, while a postnatal transition to isoform 1 is required to maintain the shorter, mechanosensitive stereocilia rows.

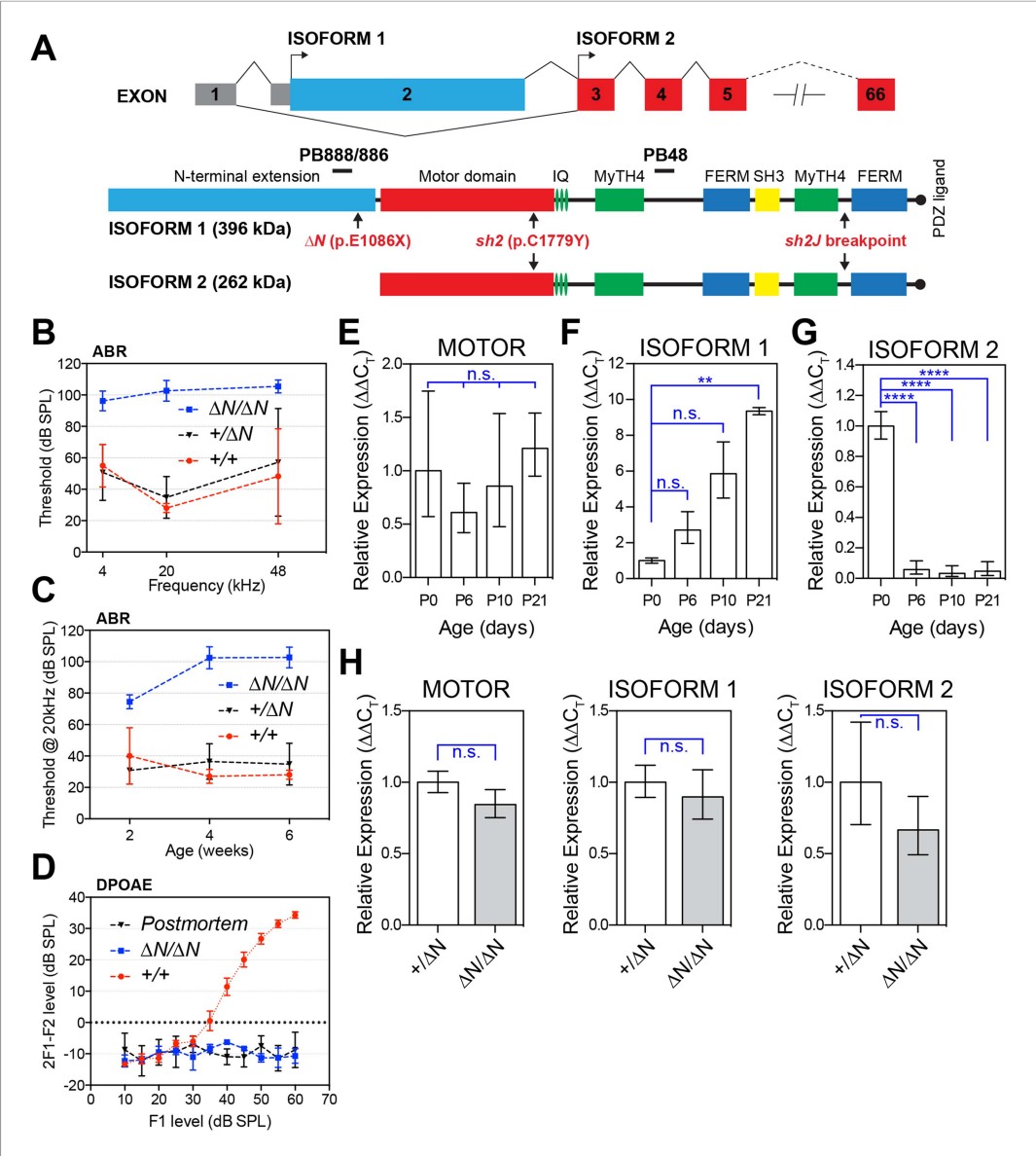

**Figure 1.** A mutation targeting isoform 1 causes deafness in $Myo15^{\Delta N/\Delta N}$ mice. (**A**) Two protein isoforms are generated from alternatively spliced transcripts of *Myo15*. Transcripts incorporating exon 2 encode isoform 1 (396 kDa; Genbank: NM_010862.2), while exclusion of this exon produces isoform 2 (262 kDa; Genbank: NM_182698.2). Both isoforms have identical motor and tail domains, including a PDZ ligand, SH3, MyTH4 and FERM moieties. The mutant *Myo15* alleles used in this study are shown along with antibody epitopes. (**B**) Auditory brainstem response (ABR) thresholds at 4, 20 and 48 kHz for $Myo15^{+/+}$, $Myo15^{+/\Delta N}$ and $Myo15^{\Delta N/\Delta N}$ mice at 6 weeks of age. Data are mean ± SD ($n = 3$–6 animals per group). (**C**) ABR thresholds at 20 kHz, measured from 2, 4 and 6 weeks old $Myo15^{+/+}$, $Myo15^{+/\Delta N}$ and $Myo15^{\Delta N/\Delta N}$ mice. Data are mean ± SD ($n = 3$–6 animals per group). (**D**) Distortion product otoacoustic emission (DPOAE) levels (2F1-F2) at 12 kHz in $Myo15^{+/+}$ and $Myo15^{\Delta N/\Delta N}$ mice at 6 weeks. Data are mean ± SD ($n = 3$–4 animals per group). (**E–G**) Relative expression of *Myo15* isoforms in wild-type cochleae measured with RT-qPCR at ages indicated. Probes target exon junction 13–14, detecting the motor domain common to both isoforms 1 and 2 (**E**); exon junction 2–3, detecting isoform 1 (**F**); exon junction 1–3, detecting isoform 2 (**G**). Relative expression $2^{(-\Delta\Delta C_T)}$ for each *Myo15* transcript was normalized first to the housekeeping gene (*Tbp*) and then to the respective isoform expression at P0. The total expression of both isoforms remains stable (**E**), however there is a transition from isoform 2 to isoform 1, which becomes the dominant mRNA species by P21 (see *Figure 1—figure supplement 1D*). Data are mean ± SD ($n = 3$–5 biological replicates per condition). Asterisks indicate significance: n.s., $p > 0.05$; **, $p < 0.01$; ****, $p < 0.0001$ (ANOVA with Tukey's multiple comparison test). (**H**) Identical qPCR probes were used to assay *Myo15* expression in $Myo15^{+/\Delta N}$ and $Myo15^{\Delta N/\Delta N}$ cochleae at P0. Relative expression $2^{(-\Delta\Delta C_T)}$ values were normalized to *Tbp* and then to expression in

*Figure 1. continued on next page*

*Figure 1. Continued*

heterozygous $Myo15^{+/\Delta N}$ samples. Data are mean ± SD (n = 3–4 biological replicates per condition). n.s., p > 0.05 (*t*-test of independent variables).

The following figure supplement is available for figure 1:

**Figure supplement 1**. Generation of a mouse model for human p.E1105X DFNB3 deafness.

## Results

### $Myo15^{\Delta N/\Delta N}$ mice are deaf

To selectively disrupt myosin 15 isoform 1 without altering the coding sequence of isoform 2, we used homologous recombination in mouse embryonic stem (ES) cells to knock-in a p.E1086X nonsense mutation into exon 2 (*Figure 1A* and *Figure 1—figure supplement 1*), mimicking the p.E1105X DFNB3 allele that causes hearing loss in humans (*Nal et al., 2007*). Because isoform 2 transcripts skip exon 2, we hypothesized that the p.E1086X mutation (referred to as $Myo15^{\Delta N}$) would specifically disrupt isoform 1. Auditory brainstem response (ABR) testing was used to measure the hearing thresholds of 6 week old $Myo15^{\Delta N/\Delta N}$ mice and their littermates at 4, 20 and 48 kHz (*Figure 1B*). $Myo15^{\Delta N/\Delta N}$ mice were profoundly deaf at all frequencies tested (*Figure 1B*). However, around the onset of hearing at 2 weeks, $Myo15^{\Delta N/\Delta N}$ mice did respond to loud sounds of 75 dB of sound pressure level (dB SPL) at 20 kHz, the most sensitive frequency range of mouse hearing (*Figure 1C*). However, by 4 and 6 weeks of age ABR thresholds at 20 kHz exceeded 100 dB SPL in $Myo15^{\Delta N/\Delta N}$ mice, indicating a rapid progression to profound deafness (*Figure 1C*). In control $Myo15^{+/+}$ and $Myo15^{+/\Delta N}$ littermates, the average thresholds measured at 20 kHz were between 27 and 40 dB SPL and did not change significantly with age (*Figure 1C*). Distortion product otoacoustic emissions (DPOAEs) were collected to evaluate active cochlear amplification by outer hair cells (OHCs). There was a complete absence of DPOAEs in $Myo15^{\Delta N/\Delta N}$ mice at 2 weeks (data not shown) and 6 weeks of age (*Figure 1D*), where in contrast $Myo15^{+/+}$ littermates had normal DPOAEs at 2 weeks (data not shown) and 6 weeks (*Figure 1D*). We conclude that cochlear amplification is disrupted in $Myo15^{\Delta N/\Delta N}$ mice, and that this contributes to the profound deafness evident by 4 weeks of age.

$Myo15^{sh2/sh2}$ and $Myo15^{sh2-J/sh2-J}$ mice have profound congenital deafness and vestibular dysfunction typified by persistent head bobbing and circling behaviors (*Probst et al., 1998*; *Anderson et al., 2000*). The residual hearing function in 2 week old $Myo15^{\Delta N/\Delta N}$ mice hinted that the hearing impairment caused by the disruption of exon 2 had a different underlying pathophysiology to these previously reported *Myo15* mouse models. This interpretation was supported by functional studies of the vestibular system. $Myo15^{\Delta N/\Delta N}$ mice lacked circling behavior and performed normally in swimming tests (data not shown). Thus, $Myo15^{\Delta N/\Delta N}$ mice differ from the $Myo15^{sh2/sh2}$ and $Myo15^{sh2-J/sh2-J}$ models in the onset and severity of deafness and also by the absence of an overt vestibular pathology.

### *Myo15* transcripts exhibit developmentally regulated isoform switching

We hypothesized that the varying severity of sensory pathology evident in *Myo15* mutant mice was due to isoforms 1 and 2 being differentially targeted. The $Myo15^{sh2}$ and $Myo15^{sh2-J}$ alleles contain a mutation in the motor domain (p.C1779Y) or a genomic deletion of exons encoding the C-terminal FERM domain and PDZ ligand, respectively (*Probst et al., 1998*; *Anderson et al., 2000*); both are predicted to ablate isoforms 1 and 2 (*Figure 1A*). In contrast, the p.E1086X mutation in exon 2 is expected to selectively target isoform 1, but preserve isoform 2 (*Figure 1A*). To investigate the expression of *Myo15* isoforms during cochlea development and maturation, we used quantitative PCR (qPCR) to measure their relative abundance in total RNA at P0 through P21. Total RNA was used as *Myo15* is primarily expressed in hair cells (*Anderson et al., 2000*; *Caberlotto et al., 2011*; *Shen et al., 2015*). Probe sets were designed to detect mRNA splicing between exons 2 and 3 (representing isoform 1), exons 1 and 3 (isoform 2), and of exons 13 and 14 within the ATPase motor domain, which independently reported the total pool of isoform 1 and 2 transcripts. The total pool of *Myo15* transcripts remained stable between P0 and P21 (*Figure 1E*), but this concealed an underlying switch in exon 2 splicing. Transcripts encoding isoform 1 became progressively more abundant, increasing

~nine-fold from P0 through to P21 (*Figure 1F* and *Figure 1—figure supplement 1D*). Conversely, the amount of isoform 2 sharply decreased ~21-fold between P0 and P6 and remained stable thereafter (*Figure 1G* and *Figure 1—figure supplement 1D*).

We used the same qPCR assays to examine the effect of the $Myo15^{\Delta N}$ allele upon isoform-specific transcript levels. There was no statistically significant difference in the quantity of isoform 1, isoform 2 or the total pool of transcripts in control $Myo15^{+/\Delta N}$ vs $Myo15^{\Delta N/\Delta N}$ cochleae at P0 (*Figure 1H*) or at P7 (data not shown). These data show that the p.E1086X mutation did not induce nonsense mediated mRNA decay or change the relative abundance of isoform transcripts. We conclude that isoform 2 mRNA was predominantly expressed during early neonatal development, but that alternative splicing progressively shifts to favor isoform 1 in postnatal cochleae.

## $Myo15^{\Delta N/\Delta N}$ hair cells initially develop a normal stereocilia architecture

To investigate the underlying pathophysiology causing deafness in $Myo15^{\Delta N/\Delta N}$ mice, we examined cochlear hair cells using scanning electron microscopy (SEM). Remarkably, unlike the short stereocilia bundles of $Myo15^{sh2/sh2}$ and $Myo15^{sh2-J/sh2-J}$ hair cells (*Probst et al., 1998*; *Anderson et al., 2000*), we found that stereocilia bundles of $Myo15^{\Delta N/\Delta N}$ inner hair cells (IHCs) had the characteristic staircase architecture and were indistinguishable from normal hearing $Myo15^{+/\Delta N}$ littermate controls at P4 (*Figure 2A*). We quantified the architecture of $Myo15^{\Delta N/\Delta N}$ and control $Myo15^{+/\Delta N}$ IHC bundles at P4 and found no differences in either the distribution of heights (*Figure 7—figure supplement 2*), or the aspect ratio of the second row mechanosensitive stereocilia tips (*Figure 7—figure supplement 3*). Tip-links were present in both $Myo15^{\Delta N/\Delta N}$ and $Myo15^{+/\Delta N}$ IHCs (*Figure 2A*, lower panels). The normal morphology of $Myo15^{\Delta N/\Delta N}$ IHC hair bundles at P4 was striking compared to those of age-matched

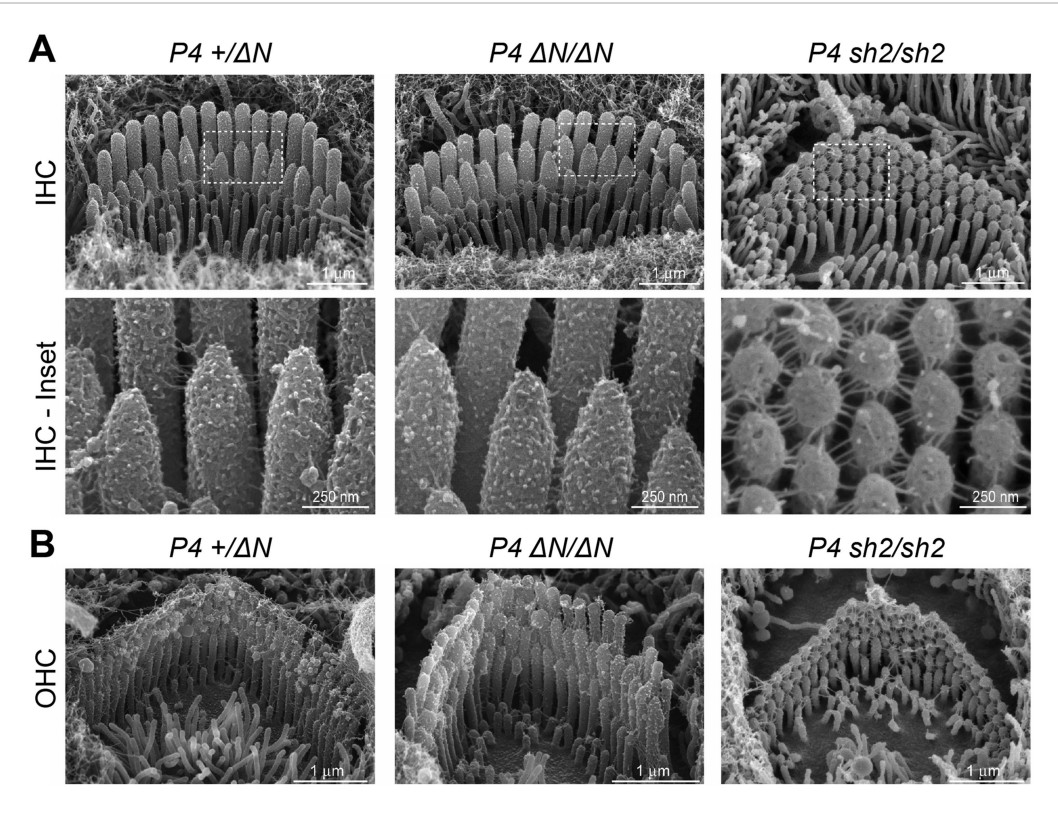

**Figure 2**. $Myo15^{\Delta N/\Delta N}$ hair cells initially develop normal stereocilia bundles. (**A**, **B**) Scanning electron microscopy (SEM) of inner (**A**) and outer (**B**) hair cells from $Myo15^{\Delta N/\Delta N}$, normal hearing $Myo15^{+/\Delta N}$ littermates and $Myo15^{sh2/sh2}$ cochleae at P4. Highlighted regions of IHC bundles (white boxes) are shown at higher magnification below. $Myo15^{\Delta N/\Delta N}$ stereocilia bundles develop the characteristic staircase architecture (see *Figure 7—figure supplement 2*) that is strikingly absent from age-matched $Myo15^{sh2/sh2}$ hair cells. Scale bars are 1 μm (**A**, upper row and **B**) and 250 nm (**A**, lower row).

$Myo15^{sh2/sh2}$ mice that were abnormally short and had immature, omnidirectional links (*Figure 2A*). Hair bundles of $Myo15^{sh2/sh2}$ IHCs also had supernumerary stereocilia rows, which were not observed in $Myo15^{\Delta N/\Delta N}$ IHCs (*Figure 2A*). Similar to IHCs, $Myo15^{\Delta N/\Delta N}$ OHCs at P4 also developed a normal hair bundle morphology that was clearly distinct from age-matched $Myo15^{sh2/sh2}$ OHCs (*Figure 2B*). Although the hair bundles of IHCs and OHCs initially developed normally at P4, they subsequently degenerated (Figure 6), and this likely contributes to the deafness observed in $Myo15^{\Delta N/\Delta N}$ mice at P14 and older (*Figure 1C*).

We conclude that the p.E1086X (*ΔN*) mutation in exon 2 causes deafness through a mechanism that is fundamentally different to the previously reported *Myo15* alleles that interfere with stereocilia elongation. The initial formation of normal stereocilia in p.E1086X mutants suggests that isoform 2 is sufficient for normal hair bundle development, consistent with our finding that the dominant *Myo15* transcript in developing P0 hair cells encodes isoform 2. Conversely, the postnatal switch to isoform 1 expression points to a distinct function for myosin 15 that is also essential for normal hearing.

## Isoform 1 localizes to the tips of shorter row mechanotransducing stereocilia

Myosin 15 has been detected at the tips of all stereocilia rows using antibodies raised against common epitopes present in both isoforms 1 and 2 (*Belyantseva et al., 2003*). To determine the localization of isoform 1 specifically, we developed antibodies PB888 and PB886 to an epitope uniquely encoded by exon 2 (*Figure 1A*). Labeling of P14 IHC bundles with PB888 (*Figure 3A*) and PB886 (data not shown) revealed that isoform 1 was concentrated at the tips of the shorter second and third row stereocilia. Isoform 1 was inconsistently detected at the tips of the tallest row, with (data not shown), or without antigen retrieval (*Figure 3—figure supplement 1F*), indicating that the asymmetric distribution in stereocilia rows was unlikely due to epitope masking. We used transmission immuno-gold electron microscopy (immuno-TEM) to study the localization of isoform 1 at higher spatial resolution. Gold particles were concentrated within the prolate tips of wild-type shorter stereocilia of P16 IHCs (*Figure 3C,D*) and were infrequently observed along the stereocilia core (*Figure 3C*). Consistent with this, isoform 1 was not detected at the upper tip-link insertion site on the tallest row in mechanically splayed IHC bundles (*Figure 3F*). In OHCs at P14, isoform 1 was similarly detected at the tips of shorter stereocilia (*Figure 3B,E*) but additionally at the tips of the tallest row (*Figure 3B,B′*). We conclude that isoform 1 localizes to the tip density of shorter row stereocilia in close proximity to the site of MET (*Beurg et al., 2009*).

Given the developmental regulation of *Myo15* splicing, we examined the localization of isoform 1 in cochleae at different postnatal ages. Similar localization patterns of isoform 1 were observed not only at P14 (*Figure 3A,B*) but also from P7 through to P28 in both IHCs and OHCs (*Figure 3—figure supplement 1C,E*), indicating that this was the distribution of isoform 1 in mature hair cells. Only trace PB888 labeling was detected in stereocilia of hair cells at P1 (*Figure 3—figure supplement 1A*), consistent with the relatively low abundance of isoform 1 transcripts detected at this age by qPCR (*Figure 1F* and *Figure 1—figure supplement 1D*). The apparent absence of isoform 1 in young hair bundles provides an explanation for why mutations of exon 2 do not affect stereocilia elongation or establishment of the hair bundle architecture.

To test the specificity of immuno-labeling for isoform 1, PB888 was examined in mutant $Myo15^{\Delta N/\Delta N}$ cochleae at P1, P7 and P14 (*Figure 3G,H*, *Figure 3—figure supplement 1B,D*) and at P14 for PB886 (data not shown). We observed no stereocilia labeling with either PB888 or PB886, confirming antibody specificity and also that the $Myo15^{\Delta N}$ allele resulted in the loss of isoform 1 from the hair bundle. Since the premature stop codon (p.E1086X) did not trigger nonsense-mediated mRNA decay in $Myo15^{\Delta N/\Delta N}$ cochleae (*Figure 1H*), we hypothesized that a truncated isoform 1 could not localize to stereocilia without the motor and tail domains. Consistent with this, PB888 labeling was absent from stereocilia of $Myo15^{sh2/sh2}$ hair cells (*Figure 3I*), indicating that the actin-binding motor domain is critical to actively localize isoform 1 within stereocilia.

## Myosin 15 isoforms are differentially trafficked within the hair bundle

Having discovered that isoform 1 was restricted to shorter row IHC stereocilia, we hypothesized that previous reports of pan-specific myosin 15 immunolabeling on the tallest row must represent isoform 2 (*Belyantseva et al., 2003*; *Rzadzinska et al., 2004*). There is no unique epitope to generate isoform

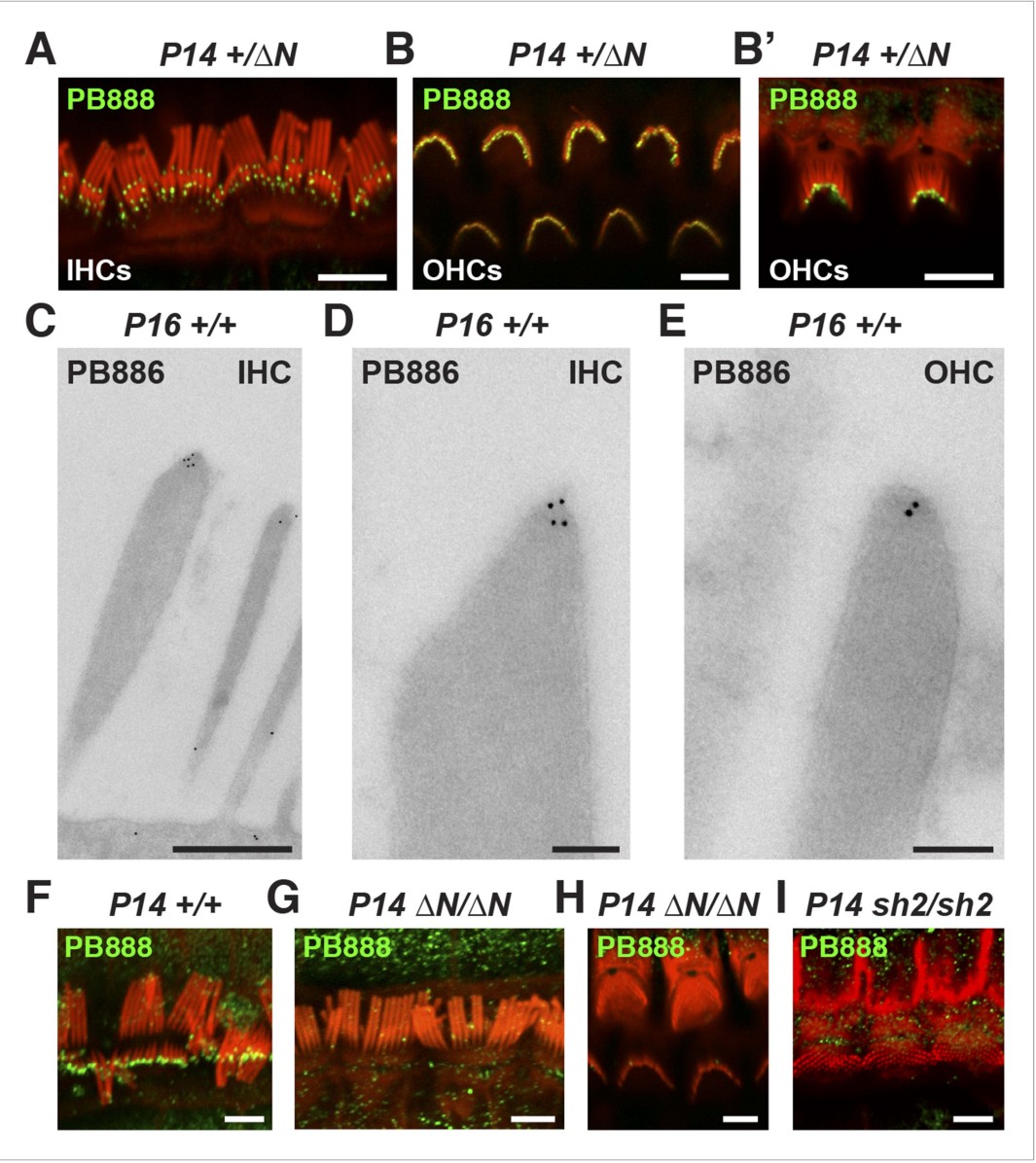

**Figure 3**. Isoform 1 targets to the tips of shorter mechanotransducing stereocilia. (**A**) PB888 antibody (green) detects isoform 1 at the tips of shorter row stereocilia of IHCs from normal hearing *Myo15*$^{+/\Delta N}$ mice. (**B**) Isoform 1 is present at the tips of all stereocilia in OHCs of normal hearing *Myo15*$^{+/\Delta N}$ mice at P14. An oblique view confirms the presence of isoform 1 on the tallest row (**B'**). (**C–E**) TEM micrographs of immuno-gold labeled PB886 in ultrathin stereocilia sections. Isoform 1 is localized in proximity to the stereocilia tip density. Labeling was infrequently observed along the stereocilia core. (**F**) PB888 does not localize to the upper tip-link insertion point on the tallest IHC stereocilia row in intentionally splayed bundles. (**G, H**) Loss of reactivity in *Myo15*$^{\Delta N/\Delta N}$ IHCs and OHCs confirms the loss of isoform 1 protein from the hair bundle and the specificity of PB888 labeling. (**I**) PB888 does not label the stereocilia tips in *Myo15*$^{sh2/sh2}$ hair cells which have the p.C1779Y motor domain mutation. Scale bars are 5 μm (**A**, **B**, **B'**, **F**, **G–I**), 500 nm (**C**), 100 nm (**D**, **E**). Immunofluorescence samples are counter-stained with rhodamine phalloidin (red) to reveal the stereocilia actin cytoskeleton. See also *Figure 3—figure supplement 1*.

The following figure supplement is available for figure 3:

**Figure supplement 1**. Isoform 1 localization during cochlear development.

2 specific antibodies (*Figure 1A*). Instead, we labeled *Myo15*$^{ΔN/ΔN}$ cochleae with an antibody raised against a common epitope present in both isoforms 1 and 2 (PB48, *Figure 1A*), reasoning that a pan-specific myosin 15 antibody should recognize only isoform 2 in isoform 1-null hair cells. In IHCs from normal controls at P7 and P14, strong PB48 labeling was detected at the tips of all stereocilia rows (*Figure 4A,B*), consistent with previous reports (*Belyantseva et al., 2003*). PB48 still strongly localized to tips of the tallest stereocilia row of littermate *Myo15*$^{ΔN/ΔN}$ IHCs, however there was a selective loss from the shorter (second and third) stereocilia rows at both P7 (*Figure 4A*) and P14 (*Figure 4B*). Because isoform 1 is absent from *Myo15*$^{ΔN/ΔN}$ IHC bundles (*Figure 3G*), we infer that PB48 was detecting isoform 2 on the tallest row. We quantified the intensity of PB48 signal on the shorter, second row IHC stereocilia relative to the tallest row. In IHCs from normal controls at P7, the intensity of PB48 labeling on the second row was 54 ± 22% (mean ± SD) of the first row signal (*Figure 4C*). In *Myo15*$^{ΔN/ΔN}$ IHCs this values was significantly reduced to 9.7 ± 5.9% of the tallest row intensity (*Figure 4C*). This indicates that the majority of myosin 15 in the second row is isoform 1 and that isoform 2 is the minor species at this location. These observations, taken

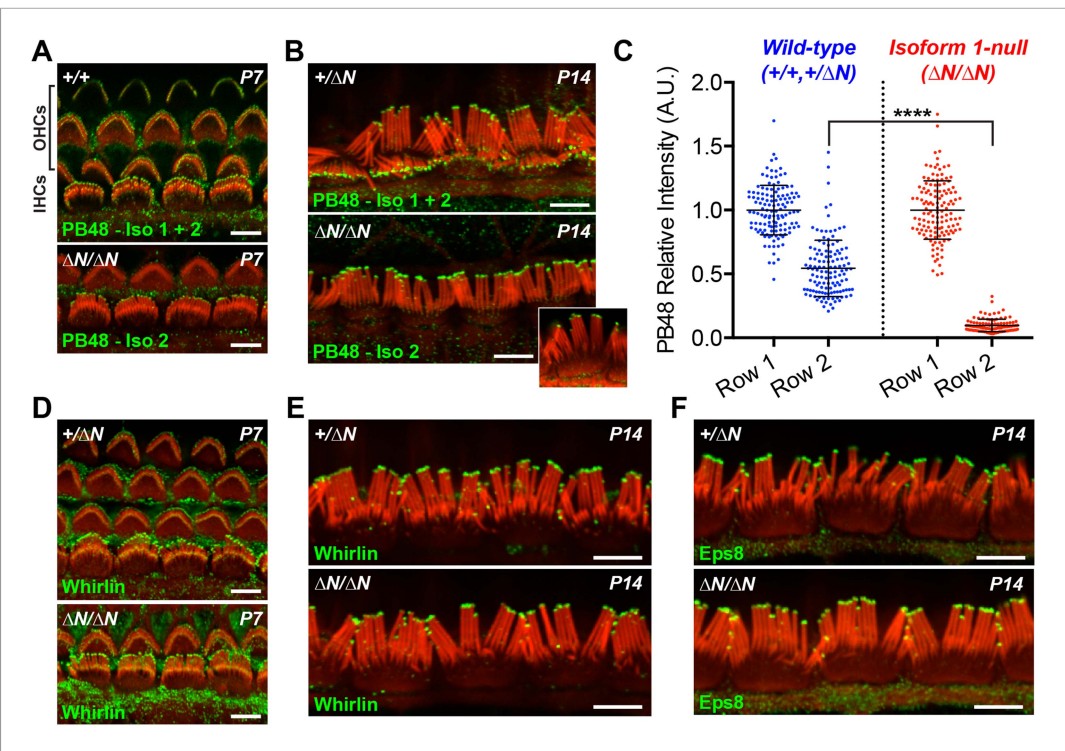

**Figure 4**. Isoform 2 traffics predominantly to the tallest stereocilia row and is sufficient to target Eps8 and Whirlin. (**A**, **B**) PB48 antibody (green) was raised to an epitope common to isoforms 1 and 2 (*Figure 1A*) and labels all stereocilia rows in wild-type *Myo15*$^{+/+}$ and *Myo15*$^{+/ΔN}$ hair cells at P7 (**A**) and P14 (**B**). In isoform 1-null *Myo15*$^{ΔN/ΔN}$ hair cells, PB48 is predominantly detected on the tallest stereocilia row at P7 (**A**) and at P14 (**B**), identifying isoform 2 at these locations. (**C**) Quantification of PB48 fluorescence on the shorter second stereocilia row of IHCs at P7 normalized to the first (tallest) row. Data points represent individual stereocilia from wild-type (blue, *Myo15*$^{+/+}$ and *Myo15*$^{+/ΔN}$ combined, n = 125 stereocilia, 3 animals) or *Myo15*$^{ΔN/ΔN}$ (red, n = 112 stereocilia, n = 4 animals) IHCs at P7, overlaid with mean ± SD. Asterisks indicate significance: *, p < 0.0001 (*t*-test of independent variables). (**D**, **E**) Whirlin antibody localizes primarily to the tallest stereocilia row of control *Myo15*$^{+/ΔN}$ hair cells at P7 (**D**) and at P14 (**E**). The localization of whirlin remains unchanged in isoform 1-null *Myo15*$^{ΔN/ΔN}$ hair cells at P7 (**D**) or P14 (**E**). (**F**) Eps8 antibody localizes primarily to the tallest stereocilia row of control *Myo15*$^{+/ΔN}$ and isoform 1-null *Myo15*$^{ΔN/ΔN}$ IHCs at P14. All samples are co-labeled with rhodamine phalloidin (red). Scale bars are 5 µm.

The following figure supplement is available for figure 4:

**Figure supplement 1**. Isoform 2 is sufficient to traffic Eps8 and Whirlin within the hair bundle.

together with PB888 data, support the conclusion that myosin 15 isoforms are sorted into different stereocilia rows as early as P7 in IHCs.

At P1, a strong PB48 signal was observed at the stereocilia tips of both IHCs and OHCs of isoform 1-null $Myo15^{\Delta N/\Delta N}$ mice that was comparable with the PB48 signal in control mice of the same age (*Figure 4—figure supplement 1A,B*). This is consistent with the early expression of isoform 2 detected by qPCR (*Figure 1G*) and its sufficient role in normal development of the hair bundle (*Figure 2A,B*). Later in development, PB48 labeling persisted at the tallest row of $Myo15^{\Delta N/\Delta N}$ IHC stereocilia at P7 and P14 (*Figure 4A,B*) but was progressively diminished in $Myo15^{\Delta N/\Delta N}$ OHC stereocilia at P7 and P14 (*Figure 4A* and *Figure 4—figure supplement 1C*), as compared to the labeling in control OHCs from littermates (*Figure 4A*, *Figure 4—figure supplement 1C*). These observations suggest that the overall decline of isoform 2 mRNA detected by qPCR (*Figure 1G*) may originate from a change in OHC expression.

In summary, our data show a developmental transition from isoform 2 to isoform 1 by the time the mature hair bundle architecture is almost fully developed (i.e. by P6). Furthermore, the independent segregation of myosin 15 isoforms within the hair bundle provides a mechanism for why the $Myo15^{\Delta N}$ and $Myo15^{sh2}$ alleles cause strikingly different hair bundle phenotypes. Whilst both isoforms are absent from $Myo15^{sh2/sh2}$ hair bundles, isoform 2 is still present in stereocilia of $Myo15^{\Delta N/\Delta N}$ hair cells and is sufficient to drive hair bundle elongation. It follows that isoform 1 has a function critical for hearing that is unrelated to stereocilia bundle development.

## Isoform 2 is sufficient to traffic Whirlin and Eps8 within the hair bundle

Myosin 15 regulates hair bundle development by transporting a molecular complex containing Eps8 and whirlin (*Belyantseva et al., 2005*; *Manor et al., 2011*) to sites of actin polymerization at the stereocilia tip (*Schneider et al., 2002*; *Drummond et al., 2015*). Since stereocilia elongated normally in isoform 1-null $Myo15^{\Delta N/\Delta N}$ hair cells, we investigated whether isoform 2 was sufficient to traffic whirlin and Eps8 within the developing hair bundle. In normal $Myo15^{+/\Delta N}$ cochleae, whirlin was concentrated at the tips of the tallest row of IHC and OHC stereocilia at both P7 and P14 (*Figure 4D,E* and *Figure 4—figure supplement 1D*). Weaker signal (relative to the tallest row) was also detected on the shorter stereocilia rows of IHCs at both ages (*Figure 4D,E*). When examined in isoform 1-null $Myo15^{\Delta N/\Delta N}$ cochleae at both P7 and P14, the localization of whirlin in IHCs and OHCs was indistinguishable from their $Myo15^{+/\Delta N}$ littermates (*Figure 4D,E* and *Figure 4—figure supplement 1D*). Similarly, Eps8 was detected at the tips of the tallest stereocilia row of control $Myo15^{+/\Delta N}$ IHCs at both P7 and P14 (*Figure 4F*, *Figure 4—figure supplement 1E*). Weaker Eps8 labeling was also detected on the second stereocilia row of control IHCs at P7 (*Figure 4—figure supplement 1E*), and on the tallest row of control OHC stereocilia at both P7 and P14 (*Figure 4—figure supplement 1E,F*). In isoform 1-null $Myo15^{\Delta N/\Delta N}$ cochleae, Eps8 labeling in IHCs and OHCs appeared unchanged from the wild-type littermates (*Figure 4F*, *Figure 4—figure supplement 1E,F,G*). We conclude that isoform 2 by itself is sufficient to establish the wild-type distribution of Eps8 and whirlin in IHCs and OHCs.

Whirlin and Eps8 bind to domains in the C-terminal tail of myosin 15 that are common to both isoforms 1 and 2 (*Belyantseva et al., 2005*; *Delprat et al., 2005*; *Manor et al., 2011*). It is striking therefore that the localization of whirlin and Eps8 do not depend upon isoform 1, despite the abundance of isoform 1 in hair bundles from P7 onwards. The apparent selectivity for isoform 2 explains how Eps8 and whirlin are primarily trafficked to the tallest stereocilia row and why the ablation of isoform 1 does not interfere with stereocilia development in $Myo15^{\Delta N/\Delta N}$ cochleae.

## Isoform 1 influences the deflection sensitivity of MET machinery in IHCs

To test whether isoform 1 is a critical component of the transduction machinery, MET currents were measured from $Myo15^{\Delta N/\Delta N}$ hair cells, which have normal staircase morphology and correctly oriented tip-links (*Figure 2* and *Figure 5A,B*). Whole cell MET current responses of young postnatal IHCs and OHCs (P3-4 + 3–5 days in vitro) were evoked by graded stereocilia deflections using a rigid probe (*Figure 5C,D*). Maximal MET current amplitudes were not statistically different between mutant $Myo15^{\Delta N/\Delta N}$ (n = 12) and control $Myo15^{+/\Delta N}$ IHCs (n = 10) (*Figure 5E*). However, the responses to small bundle deflections (150–300 nm) were significantly larger in mutant $Myo15^{\Delta N/\Delta N}$ IHCs, indicating an increased deflection sensitivity of the transduction apparatus in the absence of isoform 1 (*Figure 5E*). This sensitivity depends on the mechanical stiffness of a theoretical 'gating spring' that is connected

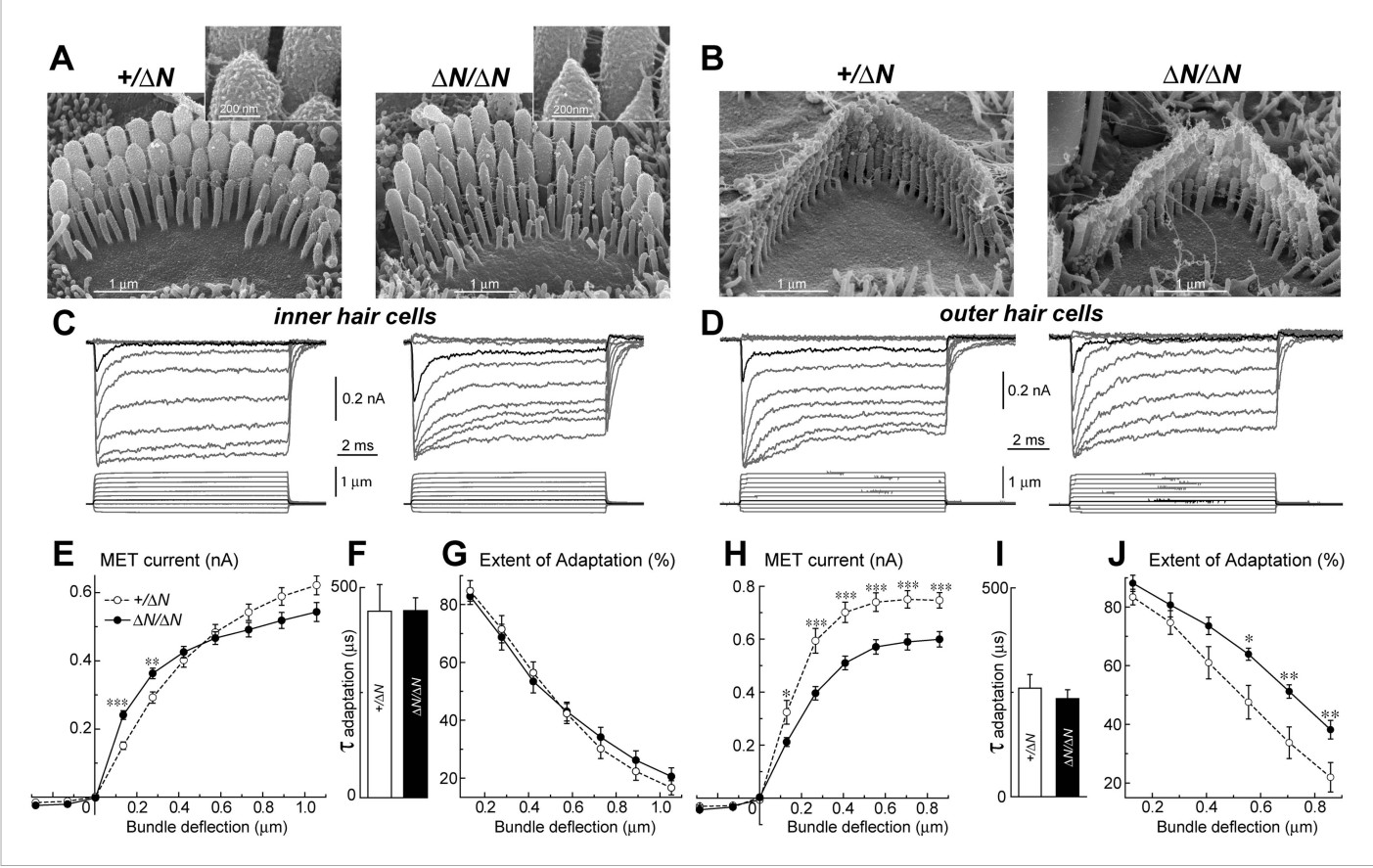

**Figure 5**. Isoform 1 is not required for MET but influences the deflection sensitivity of IHCs. (**A**, **B**) SEM images of IHC (**A**) and OHC (**B**) stereocilia bundles in *Myo15*[+/ΔN] (left panels) and *Myo15*[ΔN/ΔN] (right panels) hair cells. Higher magnification of the second row IHC stereocilia tips are shown (inset). (**C**, **D**) Whole cell current responses (top traces) evoked by graded deflections of the stereocilia bundles (bottom traces) in IHCs (**C**) and OHCs (**D**) in *Myo15*[+/ΔN] (left) and *Myo15*[ΔN/ΔN] (right) hair cells. (**E**, **H**) Relationship between the peak MET current and stereocilia bundle displacement in IHCs (**E**) and OHCs (**H**) from *Myo15*[+/ΔN] (open circles) and *Myo15*[ΔN/ΔN] (closed circles) cochleae. (**F**, **I**) Time constants of MET adaptation in IHCs (**F**) and OHCs (**I**) for *Myo15*[ΔN/ΔN] and *Myo15*[+/ΔN]. Time constants were determined from a single exponential fit of MET responses evoked by the small bundle deflections of ~150 nm (see black traces in **C**, **D**). (**G**, **J**) Percent changes of the MET current 10 ms after a stimulation step (extent of adaptation) as a function of stimulus intensity in IHCs (**G**) and OHCs (**J**). The same MET records contribute to all averaged data. Data are mean ± SE. Asterisks indicate statistical significance: *, $p < 0.01$; **, $p < 0.001$; ***, $p < 0.0001$ (*t*-test of independent variables). Holding potential was −90 mV. Age of the cells: P3-4 + 3–5 days in vitro. SEM images were obtained from cultured samples used for MET recordings. Number of cells: n = 10 (IHCs, *Myo15*[+/ΔN]), n = 12 (IHCs, *Myo15*[ΔN/ΔN]), n = 7 (OHCs, *Myo15*[+/ΔN]), n = 9 (OHCs, *Myo15*[ΔN/ΔN]).

to the MET channel (*Howard and Hudspeth, 1987*). A stiffer gating spring would transmit the maximal opening force to the MET channel at a smaller bundle deflection, resulting in earlier saturation of the current-displacement relationship. This was indeed observed in *Myo15*[ΔN/ΔN] IHCs (*Figure 5E*). Our data show that isoform 1 is not essential for MET responses in IHCs but may contribute to the stiffness of this gating spring. We did not observe a similar increase of MET sensitivity in OHCs (*Myo15*[ΔN/ΔN], n = 9; *Myo15*[+/ΔN], n = 7), perhaps due to the MET current degradation already present in *Myo15*[ΔN/ΔN] OHCs compared to *Myo15*[+/ΔN] controls (*Figure 5H*).

MET responses in both IHCs and OHCs of *Myo15*[ΔN/ΔN] mice exhibited prominent adaptation, that is, a fast decay of the MET current following stereocilia deflection (*Figure 5C,D*). The time constant of adaptation was not affected in either IHCs or OHCs of *Myo15*[ΔN/ΔN] mice compared to *Myo15*[+/ΔN] controls (*Figure 5F,I*). The extent of adaptation, represented as the percentage of the MET current decay during a step-like bundle deflection, was identical in *Myo15*[ΔN/ΔN] and *Myo15*[+/ΔN] IHCs (*Figure 5G*), and larger in *Myo15*[ΔN/ΔN] OHCs as compared to *Myo15*[+/ΔN] OHCs (*Figure 5J*). We conclude that isoform 1 is not essential for assembling the MET machinery in early postnatal development, but can contribute to the overall stiffness of the MET apparatus.

# Isoform 1 is required for postnatal maintenance of shorter row mechanotransducing stereocilia

Stereocilia and MET currents developed normally in $Myo15^{\Delta N/\Delta N}$ hair cells, however mutant mice still had severe hearing loss by P14. We investigated whether isoform 1 was essential for maintenance of the hair bundle architecture. Stereocilia ultrastructure was examined in $Myo15^{\Delta N/\Delta N}$ and control $Myo15^{+/\Delta N}$ cochlear hair cells using SEM. At P4 through P10, the gross morphology of $Myo15^{\Delta N/\Delta N}$ hair cells was similar to control $Myo15^{+/\Delta N}$ littermates (*Figure 6A,B*, top row), consistent with isoform 2 being sufficient for stereocilia development. However, in older $Myo15^{\Delta N/\Delta N}$ cochleae from P32 onwards, the mechanotransducing stereocilia in the second row were evidently reduced in height compared with normal hearing $Myo15^{+/\Delta N}$ controls. Furthermore, the third stereocilia row was almost completely resorbed by P32 in $Myo15^{\Delta N/\Delta N}$ hair cells. Remarkably, this degeneration was specific to the shorter rows that harbor active MET channels (*Beurg et al., 2009*), and it did not affect the tallest rows, at least up until P50. The same degenerative phenotype was observed in $Myo15^{\Delta N/sh2-J}$ compound heterozygotes, which produce normal isoform 2 from the $Myo15^{\Delta N}$ allele and a mutant isoform 1 lacking the tail domain from the $Myo15^{sh2-J}$ allele (*Figure 6—figure supplement 1*).

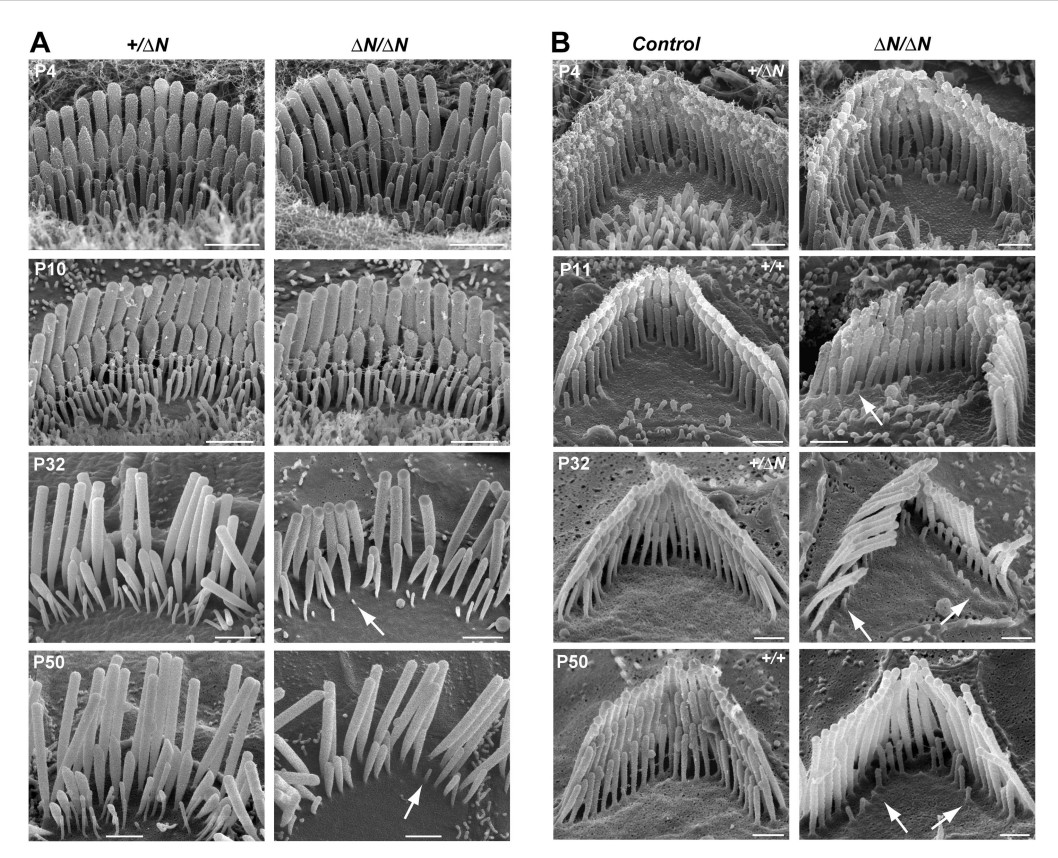

**Figure 6**. Degeneration of mechanotransducing shorter row stereocilia in isoform 1-null hair cells. (**A**) SEM micrographs of IHC stereocilia bundles of normal $Myo15^{+/\Delta N}$ (left) and $Myo15^{\Delta N/\Delta N}$ mutant (right) mice at different stages of postnatal development. Arrows point to examples of almost completely resorbed stereocilia. Note that the tallest stereocilia row does not thin, or shorten. (**B**) SEM micrographs of OHC stereocilia bundles show a similar degeneration pattern to IHCs. Shorter row stereocilia are retracted (arrows) but the tallest stereocilia row remains unaffected. All cells were located approximately at the middle of the cochlea. Scale bars are 1 μm (**A**) and 0.5 μm (**B**). See also *Figure 6—figure supplement 1* and *Table 1*.

The following figure supplement is available for figure 6:

**Figure supplement 1**. $Myo15^{sh2-J}$ does not complement $Myo15^{\Delta N}$.

These data indicate that a single genetic copy of isoform 2 is sufficient for the production of a normal hair bundle (*Table 1*) and show that isoform 1 must be full-length to maintain shorter row stereocilia.

We performed a more detailed survey of stereocilia ultrastructure at P4–P8 to understand how the degenerative process initiated in young postnatal hair cells. In the second row of $Myo15^{\Delta N/\Delta N}$ IHCs at P8, the shape of the normally prolate tips were frequently exaggerated and over-elongated (*Figure 7B*), compared to $Myo15^{+/+}$ IHCs (*Figure 7A*). We observed these over-elongated tips in $Myo15^{\Delta N/\Delta N}$ IHCs at P8, but not at P4 (*Figure 7—figure supplement 3*), suggesting this abnormality was not a developmental defect, but rather the abnormal maintenance of stereocilia in isoform 1-deficient hair cells. The over-elongated tips of $Myo15^{\Delta N/\Delta N}$ IHCs were filled with actin filaments (*Figure 7B*), indicating that actin polymerization was dysregulated at this location. The tips of stereocilia actin filaments are normally capped in an electron dense material that likely contains components of the actin polymerization machinery (*Tilney et al., 1983*). In agreement with a previous report (*Rzadzinska et al., 2004*), we confirmed that the tip density was absent from the tips of young postnatal $Myo15^{sh2/sh2}$ stereocilia (*Figure 7—figure supplement 1E–I*), supporting the role of this structure in stereocilia elongation and actin polymerization. In contrast to $Myo15^{sh2/sh2}$ stereocilia, a prominent tip density was detected at the tips of $Myo15^{\Delta N/\Delta N}$ stereocilia at P7, similar to littermate controls (*Figure 7A,B* and *Figure 7—figure supplement 1A–D*). These data are consistent with isoform 2 being necessary to form the tip density complex and drive stereocilia elongation. Whilst isoform 1 is not required for initial tip density formation, its incorporation into the postnatal structure appears to regulate actin dynamics, specifically on the shorter stereocilia rows that have active MET channels.

In further support of this idea, ablation of isoform 1 resulted in deterioration of the staircase architecture of the IHC bundle from P6 onwards (*Figure 7—figure supplement 2*). In addition, we observed a clear reduction in second row stereocilia widths measured between P8 to P11 in $Myo15^{\Delta N/\Delta N}$ IHCs (*Figure 7E*). At P8, the distribution of diameters in the second stereocilia row of $Myo15^{\Delta N/\Delta N}$ IHCs was comparable to controls (*Figure 7E*), although there was a small, statistically significant reduction in the average diameter (*Figure 7F*). By P11, the average diameter of second

**Table 1**. Hair bundle phenotypes resulting from the combination of *Myo15* alleles

| *Myo15* genotype | | Functional isoforms generated by: | | |
|---|---|---|---|---|
| Allele A | Allele B | Allele A | Allele B | Hair bundle phenotype |
| + | + | 1 and 2 | 1 and 2 | Normal |
| + | ΔN | 1 and 2 | 2 | Normal * |
| + | sh2 | 1 and 2 | - | Normal † |
| + | sh2-J | 1 and 2 | - | Normal ‡ |
| ΔN | ΔN | 2 | 2 | Normal staircase, short rows degenerate * |
| sh2-J | ΔN | - | 2 | Normal staircase, short rows degenerate * |
| sh2 | sh2 | - | - | Short staircase, additional stereocilia rows † |
| sh2-J | sh2-J | - | - | Short staircase, additional stereocilia rows ‡ |
| sh2 | sh2-J | - | - | Short staircase, additional stereocilia rows ‡ |

*Data reported in this study.
†*Probst et al. (1998)*.
‡*Anderson et al. (2000)*.
In wild-type hair cells both *Myo15* alleles can independently generate mRNA and encode protein for isoforms 1 and 2. The $Myo15^{sh2-J}$ and $Myo15^{sh2}$ alleles disrupt production of functional isoform 1 and isoform 2, whilst the $Myo15^{\Delta N}$ allele reported in this study disrupts isoform 1, but leaves isoform 2 functionally intact. Comparing different combinations of *Myo15* alleles reveals a clear genotype–phenotype correlation. Mice deficient for both isoform 1 and 2 ($Myo15^{sh2/sh2}$, $Myo15^{sh2-J/sh2-J}$, $Myo15^{sh2/sh2-J}$) have short hair bundles with additional stereocilia rows. In the presence of at least one *Myo15* allele competent to generate isoform 2 ($Myo15^{\Delta N/\Delta N}$, $Myo15^{\Delta N/sh2-J}$), the stereocilia bundle develops the normal staircase architecture, but shorter stereocilia rows degenerate postnatally. At least one functional copy of isoform 1, in addition to isoform 2 is required for normal hair bundle development and its long-term maintenance.

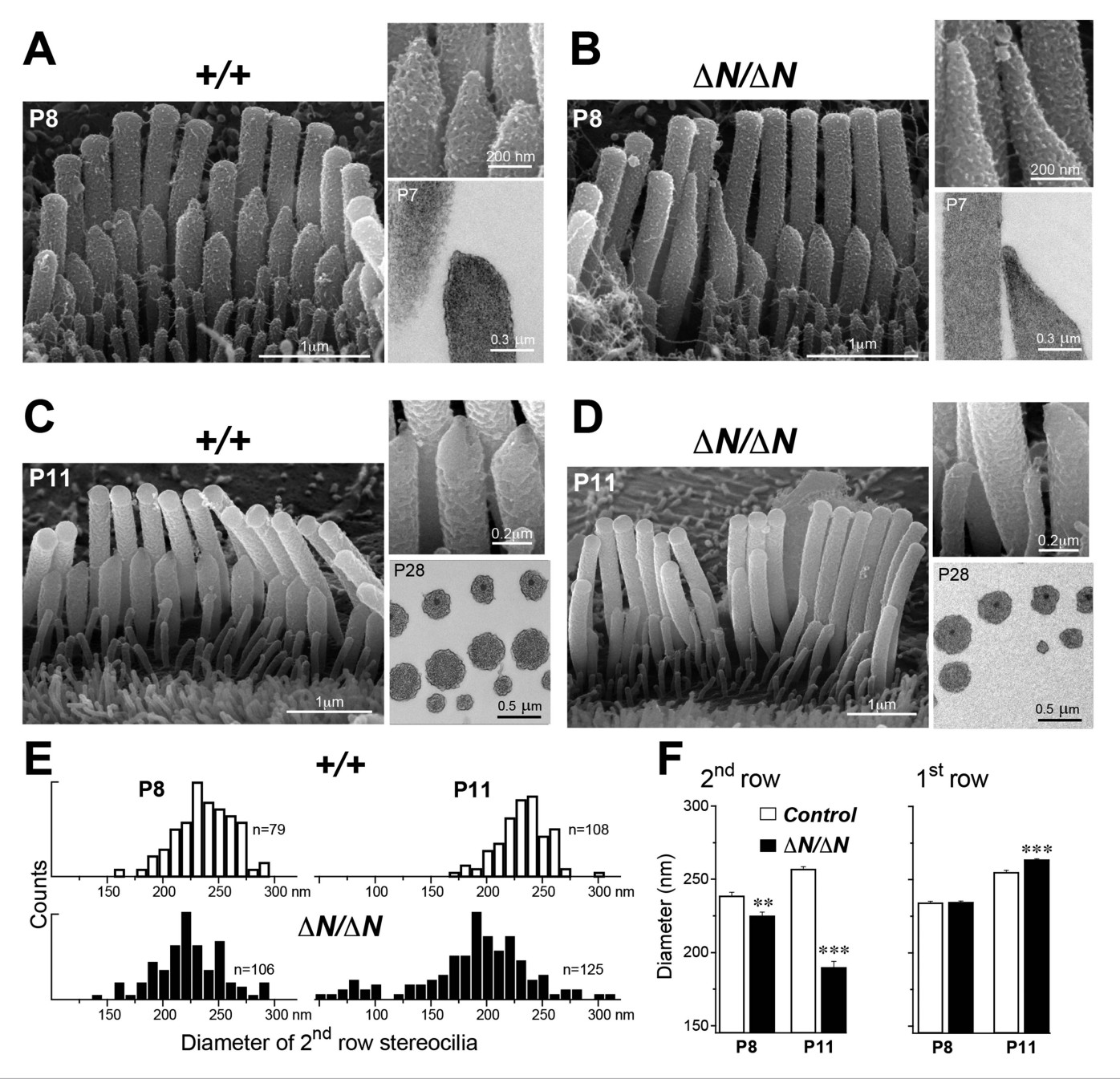

**Figure 7**. Isoform 1 maintains the diameter of mature mechanosensitive stereocilia and regulates the actin cytoskeleton at their tips. (**A–D**) SEM images of stereocilia bundles from *Myo15*$^{+/+}$ and *Myo15*$^{\Delta N/\Delta N}$ IHCs at P8 (**A**, **B**) and P11 (**C**, **D**). The inset (top right) is a higher magnification image of the second row stereocilia tips. The insets (lower right) are TEM images of either longitudinal (**A**, **B**) or axial (**C**, **D**) cross-sections of the stereocilia core. Note the electron-dense material at the tips of *Myo15*$^{\Delta N/\Delta N}$ stereocilia (**B**) and similar density of actin filaments in the thinning stereocilia of second row *Myo15*$^{\Delta N/\Delta N}$ IHCs (**D**). (**E**) Distribution of diameters of second row stereocilia in *Myo15*$^{+/+}$ (upper histogram) and *Myo15*$^{\Delta N/\Delta N}$ (lower histogram) IHCs at P8 (left), and at P11 (right). The diameters were measured from SEM images and are likely underestimated by ~30% due to the uniform tissue shrinkage during critical point drying. Number of IHCs: n = 9 (P8, *Myo15*$^{+/\Delta N}$), n = 9 (P8, *Myo15*$^{\Delta N/\Delta N}$), n = 8 (P11, *Myo15*$^{+/\Delta N}$), n = 8 (P11, *Myo15*$^{\Delta N/\Delta N}$). (**F**) Average diameter of stereocilia in the first and second rows of *Myo15*$^{+/+}$ (white bars) and *Myo15*$^{\Delta N/\Delta N}$ (black bars) IHC bundles at P8 and P11 (data from panel E). Data are shown as mean ± SE. Asterisks indicate statistical significance: **, p < 0.001; ***, p < 0.0001 (*t*-test of independent variables).

The following figure supplements are available for figure 7:

**Figure supplement 1**. Isoform 2 is necessary for tip density formation.

*Figure 7. continued on next page*

*Figure 7. Continued*

**Figure supplement 2**. Analysis of stereocilia staircase architecture in *Myo15*^ΔN/ΔN hair cells.
**Figure supplement 3**. Analysis of stereocilia tip morphology in *Myo15*^ΔN/ΔN hair cells.

row stereocilia in *Myo15*^ΔN/ΔN hair cells was significantly reduced from 256 ± 2 nm (mean ± SEM) to 189 ± 4 nm and included a distinct population of 'super-thin' stereocilia with diameters of less than 150 nm (*Figure 7E,F*). In contrast, the diameter of the tallest stereocilia row did not decrease between P8 and P11 in IHCs (*Figure 7F*). Stereocilia thinning was specific to *Myo15*^ΔN/ΔN IHCs as shorter row stereocilia in OHCs of the same mutant mice retracted without any noticeable changes in diameter (*Figure 6B*, bottom row). TEM cross-sections taken towards the base of the stereocilia core revealed a similar density of actin filaments in normal and thinned stereocilia from *Myo15*^+/+ and *Myo15*^ΔN/ΔN IHCs at P28 (*Figure 7C,D*). We infer that stereocilia thinning in *Myo15*^ΔN/ΔN IHCs was caused by the loss of core actin filaments, rather than by an increased density of actin filament packing. We conclude that isoform 1 is required for the postnatal maintenance of the actin cytoskeleton, specifically in shorter row mechanotransducing stereocilia.

## Discussion

We demonstrate that mutations in *Myo15* cause deafness through two fundamentally different mechanisms: (i) a failure to initially develop mechanosensory hair bundles, or (ii) a failure to maintain them once assembled. The involvement of myosin 15 isoform 2 in stereocilia development was previously established (*Belyantseva et al., 2005*), however our finding that isoform 1 is required to maintain mature stereocilia was unexpected. These contrasting functions for myosin 15 are underlined by their different expression profiles during development. Furthermore, the isoforms have distinct spatial localizations within the hair bundle. A growing number of proteins exhibit asymmetrical distributions within different stereocilia rows of the hair bundle (*Peng et al., 2011*), yet how these gradients are formed remains unclear. Our study shows that myosin 15 isoforms can selectively traffic to different stereocilia rows and establish the distribution of other proteins (e.g., whirlin and Eps8) critical for stereocilia development.

What molecular signposts influence the trafficking of myosin 15 isoforms to specific stereocilia rows? Because isoform 2 has no unique sequence when compared with isoform 1, the unique N-terminal extension of isoform 1 must be responsible for regulating its ultimate destination within the hair bundle. Several targeting mechanisms are conceivable. Under a selective retention model, both isoforms would uniformly enter all stereocilia, but be captured at the tips of the tallest or shorter rows by isoform-specific interactions. Alternatively, a selective entry model would specifically restrict isoforms from entering at the base of each stereocilium. A physical 'gate keeper' structure restricts cytosolic access to primary cilia (*Dishinger et al., 2010*; *Hu et al., 2010*), but an equivalent apparatus has yet to be identified in stereocilia or actin-rich protrusions like microvilli. Rather than being restricted by a physical barrier, the two myosin 15 isoforms may exhibit different motility on stereocilia actin filaments. The *shaker 2* (p.C1779Y) mutation within the actin-binding motor domain prevents both isoforms from accumulating in stereocilia, and indicates that the interaction with actin is likely critical in the trafficking process. Actin-binding proteins can direct myosin motility within cells (*Brawley and Rock, 2009*) by enhancing, or inhibiting the acto-myosin binding interface to create permissive or restrictive actin tracks. Although row-specific variations in actin topology have not yet been identified (*Tilney et al., 1983*), stereocilia and rootlets contain a plethora of actin-binding proteins that may provide guidance cues to spatially regulate the activity of myosin 15 isoforms (*Drummond et al., 2012*; *Shin et al., 2013*; *Kitajiri et al., 2010*). The recent purification of myosin 15 will allow its motility to be tested on different types of cross-linked actin filaments, and to examine how the N-terminal extension might modulate this activity (*Bird et al., 2014*; *Hartman et al., 2011*).

We found that Eps8 and whirlin do not require isoform 1 to traffic to their normal locations within the hair bundle. The sole dependence upon isoform 2 is intriguing since both myosin 15 isoforms share identical Eps8 and whirlin binding sites in the tail domain (*Liang et al., 1999*; *Belyantseva et al., 2005*; *Manor et al., 2011*). The molecular basis for this selectivity is unknown, although a distinct possibility is

that the N-terminal extension may interact with the tail domain of myosin 15 to regulate binding to cargo proteins. This type of intra-molecular regulation is common in other myosin classes (*Sellers and Knight, 2007*). We postulate that selective binding of cargo protein to isoform 1, or 2, forms the basis of a targeted protein trafficking system within mechanosensory hair bundles. Binding partners of the N-terminal extension are now of great interest, since isoform 1 may be responsible for trafficking proteins to the vicinity of the MET machinery on the shorter stereocilia rows.

The spatial segregation of myosin 15 isoforms within the hair bundle has implications for existing models of stereocilia length regulation, which remain controversial. In one model, isoform 2 is required for elongation but does not set the absolute length (*Belyantseva et al., 2003*). This model is supported by the fact that exogenous expression of isoform 2 does not cause additional elongation of developing stereocilia in vitro (*Belyantseva et al., 2003*). An alternate model posits that the quantity of myosin 15 at each stereocilia tip determines its length, and that graded increases on each row specify the overall staircase architecture (*Rzadzinska et al., 2004*; *Manor and Kachar, 2008*). We found that although isoform 1-null *Myo15*$^{\Delta N/\Delta N}$ hair cells have a reduced quantity of myosin 15 on their shorter stereocilia rows (*Figure 4C*), no gross change in the initial development of the staircase architecture was observed (*Figure 7—figure supplement 2*). These data argue against the hypothesis that myosin 15 'dose' controls the length of individual stereocilia.

Isoform 1 is dispensable for the normal elongation of stereocilia in young postnatal hair cells; by inference, we conclude that isoform 2 is sufficient to drive stereocilia development. There is growing experimental evidence that myosin 15 regulates the actin cytoskeleton in developing stereocilia: (i) *Myo15*$^{sh2/sh2}$ hair cells have short stereocilia where the actin core fails to elongate (*Probst et al., 1998*); (ii) exogenous expression of isoform 2 in *Myo15*$^{sh2/sh2}$ hair cells rescues elongation of the stereocilia core, which entails actin polymerization (*Belyantseva et al., 2005*); (iii) expression of isoform 2 in COS-7 cells induces filopodia formation, again entailing actin polymerization (*Belyantseva et al., 2003*); (iv) our present study shows that the loss of isoform 1 results in progressive disassembly of mature mechanotransducing stereocilia. Considering the exceptional stability of the stereocilia actin core (*Zhang et al., 2012*; *Drummond et al., 2015*; *Narayanan et al., 2015*), this implies changes to how the actin cytoskeleton is regulated at the stereocilia tips of isoform 1-null hair cells. Actin filaments at the stereocilia tip are embedded within an electron-dense plaque, which is proposed to cap the filament barbed ends and regulate polymerization (*Tilney et al., 1983*; *DeRosier and Tilney, 2000*; *Rzadzinska et al., 2004*). The tip density was prominent in all rows of *Myo15*$^{\Delta N/\Delta N}$ stereocilia (*Figure 7—figure supplement 1*), indicating that isoform 1 was not necessary for formation of this critical structure. However, once stereocilia are assembled, the postnatal incorporation of isoform 1 into the tip density, along with its cargoes, may repurpose the actin polymerization machinery from driving elongation towards the fine-tuning of local actin dynamics; which are normally restricted to the tips of mature stereocilia (*Zhang et al., 2012*; *Drummond et al., 2015*; *Narayanan et al., 2015*). Accordingly, we note that the onset of stereocilia abnormalities in the second row (*Figure 7—figure supplement 3*) and changes to the staircase architecture (*Figure 7—figure supplement 2*) occur around the same time as the appearance of isoform 1 within normal hair bundles (i.e. by P7).

The selective degeneration of only mechanotransducing stereocilia in *Myo15*$^{\Delta N/\Delta N}$ hair cells may result from the previously hypothesized link between MET currents and actin polymerization (*Tilney et al., 1988*). This is consistent with the tallest stereocilia rows not degenerating in *Myo15*$^{\Delta N/\Delta N}$ hair cells, since MET currents are not detected on the tallest rows (*Beurg et al., 2009*). A role for mechano-transduction has been further evidenced in hair cells of mutant *Ush1g*$^{flox/flox}$ mice, which lose tip-links and MET currents prior to degeneration of the shorter row stereocilia (*Caberlotto et al., 2011*). However, the loss of MET currents can occur alone without any apparent effects on stereocilia morphology (*Kawashima et al., 2011*). Furthermore, the absence of MET currents was unlikely the cause of stereocilia disassembly in *Myo15*$^{\Delta N/\Delta N}$ hair cells, since we found grossly normal transduction in these cells. We argue that membrane tension is instead a critical factor controlling actin polymerization at the tips of mature mechanosensitive stereocilia (*Diz-Munoz et al., 2013*; *Barr-Gillespie, 2015*) and that isoform 1 may contribute to a tension-sensing mechanism. The selective degeneration of shorter mechanotransducing stereocilia in *Myo15*$^{\Delta N/\Delta N}$ hair cells is reminiscent of the phenotype described in *Ush1g*, *Tmhs*, *Eps8L2*, *Xirp2*, *Fscn2*, *Dstn*, *Wdr1* and *Pls1* mouse mutants (*Caberlotto et al., 2011*; *Xiong et al., 2012*; *Furness et al., 2013*; *Perrin et al., 2013*; *Francis et al., 2015*; *Narayanan et al., 2015*; *Scheffer et al., 2015*; *Taylor et al., 2015*). Our observations allude to isoform 1 being part of a larger

macromolecular complex, or signaling pathway, that maintains shorter row stereocilia with active MET channels.

How might isoform 1 regulate the stereocilia actin cytoskeleton in response to tension? One possibility is that isoform 1 traffics components of a tension-sensing mechanism directly to the MET machinery. The 133-kDa N-terminal extension, unique to isoform 1, contains dense clusters of poly-proline helices (data not shown) that are ligands for SH3, WW and Enabled/VASP homology (EVH1) domains commonly found in actin-regulatory proteins. The N-terminal extension may thus interact with a broad range of proteins capable of directly orchestrating cytoskeletal dynamics. The unusual physical properties of the N-terminal extension raise another distinct possibility. The N-terminal extension is predicted as intrinsically disordered due to its high-proline content (16.8%) and low sequence complexity (http://dis.embl.de; data not shown). Intrinsically disordered domains can explore multiple structural conformations (*Dunker et al., 2002*), and in some cases act as entropic spring elements, such as the PEVK domains within the giant muscle protein, titin (*Linke et al., 1998*; *Watanabe et al., 2002*). We found that hair bundle deflection sensitivity was increased in isoform 1-null *Myo15*$^{\Delta N/\Delta N}$ IHCs, consistent with the alteration of a spring element within the MET machinery. In a further parallel with the N-terminal extension, the reversible unfolding of titin's PEVK domains during muscle extension is proposed to expose buried SH3 ligands and enable force-dependent biochemical signaling (*Ma et al., 2006*). If the N-terminus were similarly mechano-sensitive, this may result in tension-dependent interactions with SH3, WW and EVH1 domains, and retention of isoform 1 at sites of mechanical stress, e.g. in the vicinity of the lower tip-link insertion point on stereocilia. A mechano-sensitive N-terminal extension may also explain the localization of isoform 1 to the tallest rows of OHC stereocilia, since the tips likely experience tension from their attachment to the tectorial membrane during sound-induced deflections. Understanding the properties of isoform 1 and the N-terminal extension are now key to deciphering how the actin cytoskeleton is regulated in adult stereocilia, and to ultimately understand how these mechanosensory organelles are continually maintained throughout life.

## Materials and methods

### Generation of *Myo15*$^{E1086X\ (\Delta N)}$ mice

To model the c.3313 G > T (p.E1105X) allele of *MYO15A* (Genbank: NM_016239.3) that causes DFNB3 deafness in humans (*Nal et al., 2007*), we made the equivalent amino acid change c.3256GAG > TAA (p.E1086X) in *Myo15* (Genbank: NM_010862.2) using homologous recombination in mouse ES cells (*Figure 1—figure supplement 1A*). Homologous recombination arms encompassing the genomic sequences of *Myo15* exon 2 (5′ arm), and exons 3 through exon 7 (3′ arm), were amplified from 129X1/SvJ genomic DNA using polymerase chain reaction (PCR), and ligated into a pflox plasmid (*Chui et al., 1997*) previously modified to remove the HSVtk cassette. The c.3256GAG > TAA mutation was introduced using the QuickChange II XL Site-Directed Mutagenesis Kit (Agilent, Santa Clara, CA). This created a new *Mse*I site that was used for genotyping. An additional *Asp*718 site was engineered adjacent to the *loxP* site in intron 2 to aid Southern analysis. The targeting vector was fully sequenced on both strands. Correctly recombined ES cells were screened by Southern blot analysis (*Figure 1—figure supplement 1B*) and injected into C57BL/6J blastocysts. Positive F1 progeny were crossed with EIIA–Cre mice (*Lakso et al., 1996*) to remove the neomycin-resistance cassette. Except for a single residual *loxP* site and *Asp*718 site in non-conserved regions of intron 2, the resulting B6.129P2-*Myo15*$^{(E1086X/E1086X)}$ mouse strain (referred to as *Myo15*$^{\Delta N/\Delta N}$) had a structurally intact *Myo15* genomic locus. Mouse tail biopsies were genotyped by PCR (primers 5′-CCACAGTCTGAGGACCGAGT-3′ and 5′ GGTCTTGGTCTGGATGCTCT-3′). The resulting amplicon was analyzed by *Mse*I restriction endonuclease digestion; the *Myo15*$^+$ allele generates 445 bp and 30 bp restriction fragments, whereas the *Myo15*$^{\Delta N}$ allele generates 324 bp, 121 bp and 30 bp products (*Figure 1—figure supplement 1C*).

Shaker 2 (*Myo15*$^{sh2/sh2}$), shaker 2J (*Myo15*$^{sh2-J/sh2-J}$), EIIa-Cre and C57BL/6J mice were obtained from the Jackson Laboratory (Bar Harbor, ME, USA). All animal procedures were approved by the Animal Use and Care Committees (ACUC) at the University of Michigan (#PRO00004639, #PRO00005913, #PRO00005128), the University of Kentucky (#903M2005) and the NIDCD/NIH (#1263-12).

## Assessment of hearing and outer hair cell function

ABRs and DPOAEs were recorded and analyzed as described previously (*Karolyi et al., 2007*). Mice were tested at 4, 20 and 48 kHz for ABRs (n = 3–6 mice per condition) and 12, 24, and 48 kHz for DPOAEs (n = 3–4 mice per condition).

## RNA extraction and quantitative real-time PCR

The otic capsule and vestibular labyrinth were dissected in RNAlater (Life Technologies, Frederick, MD) and total RNA isolated from cochlear tissues using RNAqueous-4PCR (Life Technologies). First strand cDNA was generated using random-primed SuperScript III First-Strand Synthesis System for RT-PCR (Life Technologies). TaqMan assay IDs (Life Technologies) #Mm00465026_m1 (*Myo15* exon 13–14), #Mm04205306_m1 (*Myo15* exon 2–3) and #AJMSGCH (*Myo15* exon 1–3) were run on a Real-Time PCR System (ABI7500, Life Technologies). Transcripts for TATA-binding protein (*Tbp;* #Mm00446971_m1) were used as the housekeeping reference gene. All reactions were performed in triplicate and averaged. A minimum of 3 independent biological replicates (from different animals) was performed for each age group and/or genotype. The difference in cycle threshold between *Myo15* and *Tbp* was calculated for each sample ($\Delta C_T$), and then normalized ($\Delta\Delta C_T$) to either P0 (*Figure 1E–G*), or $Myo15^{+/\Delta N}$ (*Figure 1H*) samples. Relative expression was calculated as $2\char94(-\Delta\Delta C_T)$.

## Immunofluorescence

PB886 and PB888 antisera were generated by immunizing New Zealand White rabbits (Covance, Denver, PA) with a KLH-coupled peptide [H]-CKKFLGQHHDPGPGQLTKSAD-[NH$_2$] (Princeton Biomolecules Corp, NJ). Antisera were peptide affinity purified before use. Immunofluorescence protocols were performed as described (*Belyantseva et al., 2003*). Briefly, mouse cochleae were fixed with 4% paraformaldehyde in PBS for 2 hr at room temperature. Dissected cochleae were permeabilized in 0.5% Triton X-100 in PBS for 30 min. An optional 30 min incubation in 0.1M sodium citrate (pH 6) at 60C was included for antigen retrieval. Samples were blocked in 5% NGS/2% BSA in PBS and incubated in primary antibodies at 4C: PB888/886, Eps8 (#610143, BD Biosciences, San Jose, CA), PB48 (*Liang et al., 1999*) or HL5136 (*Belyantseva et al., 2005*). Primary antibodies were detected using Alexa Fluor 488 conjugated secondary antibodies (Life Technologies). Samples were labeled with rhodamine-phalloidin (Life Technologies) and mounted in Prolong Gold (Life Technologies), before imaging with a LSM780 confocal microscope (Zeiss, Thornwood, NY) and a 63x oil objective (Plan-Apochromat,1.4 N.A.).

## Fluorescence quantification and statistical analyses

Confocal z-stacks of P7 IHCs labeled with PB48 were analyzed using ImageJ (http://imagej.nih.gov). For each hair cell, ~10 circular regions of interest (ROI; 500 nm diameter) were placed at the tips of the tallest stereocilia (first row) and additionally ~10 more (400 nm diameter) at the tips of the second row. The z-stack position was adjusted for each individual ROI to obtain the maximum fluorescence signal. Integrated fluorescence values were divided by the corresponding ROI area to yield an integrated fluorescence density for each stereocilia tip. To allow comparisons between stereocilia from different samples, fluorescence densities of individual ROIs (from the first and second row) from a single hair cell were normalized to the mean fluorescence density of the first row measured from the same hair cell. This yielded a relative intensity (RI) index, that expresses the fluorescence density at each stereocilia tip as a ratio of the fluorescence intensities observed (on average) at the tips of the tallest stereocilia row.

Implicit in this analysis is the assumption that fluorescence values on the tallest stereocilia row remain constant between different genotypes, and can thus act as an internal calibration standard. Although PB48 fluorescence intensities at the tallest stereocilia row were subjectively comparable between $Myo15^{+/+}$ and $Myo15^{\Delta N/\Delta N}$ IHCs, we cannot exclude the possibility of a systematic difference. We stress that the RI index does not allow for comparisons of absolute fluorescence between different genotypes, but only of relative changes.

All statistical analyses (*t*-test and ANOVA) were performed using Prism v6.0 (GraphPad, La Jolla, CA) with two-tailed p-values reported.

## Scanning electron microscopy and quantification

Cochleae were perfused with 2.5% glutaraldehyde (Electron Microscopy Sciences, Hatfield, PA) in 0.1M sodium cacodylate (pH 7.4) with 2 mM $CaCl_2$ for 1–2 hr at room temperature, and then micro-dissected in distilled water. Specimens were dehydrated in a graded ethanol series, critical-point dried in liquid $CO_2$, mounted on stubs using double-stick carbon tape and sputter-coated with 4–5 nm of platinum (Q150T, Quorum Technologies, United Kingdom). Samples were examined using a field-emission SEM (S-4300 or S-4800, Hitachi, Japan).

The stereocilia staircase architecture was quantified using a previously published approach (*Xiong et al., 2012*). Briefly, direct tip-to-tip measurements (in nm) were made from the tip of the tallest stereocilia row (first row) to the tip of the second row, and from the tip of the second row to the tip of the third row. Additionally, the height of the third stereocilia row was measured relative to its insertion into the apical surface of the hair cell. Hair bundles were imaged as perpendicular to the stereocilia axis as possible to minimize projection errors. To quantify the shape of the second row stereocilia tips in IHCs, an outline was traced around the edge of each stereocilium, extending up to 300 nm down from the most distal point of the tip. An ellipse was fit to this region of interest using ImageJ (http://imagej.nih.gov/ij/) and the aspect ratio calculated as the ellipse major axis divided by the ellipse minor axis. Both types of morphological analyses were made blinded to genotype.

## Transmission electron microscopy

Cochleae were fixed in 2.5% glutaraldehyde and 2% paraformaldehyde in 0.1 M cacodylate buffer overnight at 4C, and then micro-dissected in PBS, followed by 3 × 20 min washes in 0.1 M cacodylate buffer (pH 7.2). Samples were post-fixed for 30 min in 1% osmium tetroxide, washed again in cacodylate buffer, dehydrated in series of 25%, 35% and 50% ethanol, then stained with 1% uranyl acetate in 50% ethanol and dehydrated further in 75% and 100% ethanol. Samples were embedded in Epon (PolyBed 812, Polysciences Inc, Warrington, PA) and 60–80 nm thin sections cut using an Ultracut UCT (Leica, Buffalo Grove, IL), collected on copper grids, post-stained with uranyl acetate and lead citrate, and imaged using a JEM 1010 electron microscope (JEOL, Japan).

## Immunogold electron microscopy

Post-embedding immunogold labeling was performed as described (*Petralia and Wenthold, 1999*) with minor modifications. Cochleae were perfused with 4% paraformaldehyde and 0.25% glutaraldehyde in PBS, cryoprotected in 30% sucrose, freeze-substituted and embedded in Lowicryl HM-20 resin (Electron Microscopy Sciences). Ultrathin sections were cut using an Ultracut UCT (Leica), collected on nickel grids and treated with 0.1% sodium borohydride with 50 mM glycine in TBST, incubated in 10% NGS in TBST, followed by overnight incubation at 4C in primary antibody (PB886) diluted in 1% NGS/TBST. After washing in TBST, grids were blocked in 1% NGS/TBST for 10 min followed by incubation using 1:20 dilution of goat $F(ab)_2$ anti-rabbit IgG conjugated to 10 nm gold particles (Ted Pella Inc., Redding, CA) in 1% NGS and 0.5% polyethylene glycol (20,000 MW) in TBST for 1 hr at room temperature. Finally, sections were stained with 1% uranyl acetate and 0.3% lead citrate and examined using a JSM-1010 TEM microscope (JEOL).

## Whole-cell patch clamp recordings

Organ of Corti explants were dissected at P3-P4 and cultured in glass bottomed WillCo Wells (Chemglass, Vineland, NJ) for 3–5 days in DMEM (Life Technologies) supplemented with 7% fetal bovine serum (Atlanta Biologicals, Flowery Branch, GA) and 10 mg/l ampicillin (EMD Millipore, Billerica, MA) at 37˚C (5% $CO_2$) as previously described (*Stepanyan and Frolenkov, 2009*). Experiments were performed at room temperature in Leibovitz L-15 (Life Technologies) containing the following inorganic salts (in mM): NaCl (137), KCl (5.4), $CaCl_2$ (1.26), $MgCl_2$ (1.0), $Na_2HPO_4$ (1.0), $KH_2PO_4$ (0.44), $MgSO_4$ (0.81). Hair cells were observed with a TE2000 inverted microscope (Nikon, Melville, NY) using an 100x oil-immersion objective and differential interference contrast optics. To access the basolateral plasma membrane of the hair cells, the outermost cells were removed by gentle suction with a ~5 µm micropipette. Smaller pipettes for whole-cell patch-clamp recordings were filled with intracellular solution containing (in mM): CsCl (140), $MgCl_2$ (2.5), $Na_2ATP$ (2.5), EGTA (1.0), HEPES (5.0). The pipette resistance was typically 4–6 MΩ when measured in the bath. Patch clamp

recordings were performed with an AxoPatch 200B amplifier (Molecular Devices, Sunnyvale, CA). Series resistance was compensated (up to 80%, lag 7–10 µs). After compensation, the time constant of the recording system was in the range of 35–70 µs. Hair cells were held at −60 mV except for short periods of MET recordings, when the holding potential was temporarily changed to −90 mV. All recorded hair cells were approximately at the middle turn of the cochlea.

Hair bundles were deflected using a stiff glass probe that was fire-polished to a diameter of ∼5–7 µm, matching the shape of the hair bundle. The protruding part of the probe was 2–3 mm long, which prevented lateral resonances. The probe was moved by a fast piezo actuator (PA 8/14 SG, Piezosystem Jena, Hopedale, MA), custom-modified for a faster response with a time constant of 26–28 µs. The built-in strain gauge sensor of this actuator provided a direct reading of the probe's axial displacement. The angle between the axis of the probe movement and the bottom surface of a dish was kept at ∼30˚.

## Acknowledgements

We thank Dennis Drayna, Lisa Cunningham, Katie Kindt and Melanie Barzik for critical reading; and Stacey Cole, Elizabeth Wilson, Joe Duda, Karin Halsey, Lisa Kabara, Jennifer Benson, Stephanie Edelmann, Anastasiia Nelina and Ron Petralia for expert technical assistance. This research was supported by funds from the NIDCD intramural research program DC000039-18 and DC000048-18 (JEB, IAB, TBF), NIDCD extramural funds R01 DC05053 (SAC, GIF, QF, MM, and AAI), R01 DC008861 (AAI, GIF), P30 DC05188 (DFD), the Hearing Health Foundation (MM) and a University of Michigan Barbour Scholarship and James V. Neel Fellowship (QF). We thank the University of Michigan Transgenic Animal Model Core and grants that support them (P30 CA46592), and the animal care staff at each institution.

## Additional information

### Funding

| Funder | Grant reference | Author |
| --- | --- | --- |
| National Institute on Deafness and Other Communication Disorders (NIDCD) | DC000039-18 | Thomas B Friedman, Inna A Belyantseva, Jonathan E Bird |
| National Institute on Deafness and Other Communication Disorders (NIDCD) | DC000048-18 | Thomas B Friedman, Inna A Belyantseva, Jonathan E Bird |
| Hearing Health Foundation (HHF) | N/A | Mirna Mustapha |
| University of Michigan (U-M) | N/A | Qing Fang |
| National Institute on Deafness and Other Communication Disorders (NIDCD) | DC05053 | Qing Fang, Artur A Indzhykulian, Mirna Mustapha, Gregory I Frolenkov, Sally A Camper |
| National Institute on Deafness and Other Communication Disorders (NIDCD) | DC008861 | Artur A Indzhykulian, Gregory I Frolenkov |
| National Institute on Deafness and Other Communication Disorders (NIDCD) | DC05188 | David F Dolan |

The funders had no role in study design, data collection and interpretation, or the decision to submit the work for publication.

### Author contributions

QF, AAI, GIF, JEB, Conception and design, Acquisition of data, Analysis and interpretation of data, Drafting or revising the article; MM, Conception and design, Acquisition of data, Drafting or revising the article; GPR, DFD, Acquisition of data, Drafting or revising the article; TBF, SAC, Conception and design, Analysis and interpretation of data, Drafting or revising the article; IAB, Acquisition of data, Analysis and interpretation of data, Drafting or revising the article

## Author ORCIDs

Qing Fang, http://orcid.org/0000-0002-8808-8523
Jonathan E Bird, http://orcid.org/0000-0001-5531-8794

## Ethics

Animal experimentation: All animal procedures were approved by the institutional animal care and use committees (IACUC) at the University of Michigan (#PRO00004639, #PRO00005913, #PRO00005128), the University of Kentucky (#903M2005) and at the NIDCD (#1263-12).

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
