## [Decision Letter]

Thank you for submitting your work entitled “An unique N-terminal domain enables myosin 15 to maintain mechanotransducing stereocilia and is essential for hearing” for peer review at *eLife*. Your submission has been favorably evaluated by K VijayRaghavan (Senior editor), a Reviewing editor, and three reviewers.

The following individuals responsible for the peer review of your submission have agreed to reveal their identity: Walter Marcotti (reviewer #1), the other two reviewers chose to remain anonymous.

All of the reviewers were impressed with the importance and novelty of your work. We look forward to receiving the revised version of your manuscript, letter. The three full reviews are included in this letter, as there are multiple specific and useful suggestions in them.

Reviewer #1:

In this manuscript the authors have provided an in depth analysis of the function of myo15 in the growth and maintenance of the stereociliary hair bundle of the mammalian auditory hair cells. The hair bundle is composed of an array of actin-based stereocilia of increasing height projecting from the top of hair cells. This complex structure is the site of mechanotransduction. Despite the pivotal role of the hair bundle in sound detection, we still have a very limited understanding of how it is put together during development and maintained throughout life. Myosin 15 has been shown to be required for the initial elongation of stereocilia. Hair cells express two isoforms of myosin 15 through alternative spicing (isoform 1 and 2), but the function of isoform 1 is still unknown. Isoform1 transcript includes the exon 2, which causes deafness in human when mutated. The authors have generated a knockout mouse to ablate isoform 1 but leaving the other isoform intact. They have found a time-dependent expression of both isoforms with 2 being mainly expressed at very early stages of development (hair bundle elongation period) and 1 during later stages (maintenance period). Moreover, the two isoforms appear to selectively traffic to different stereocilia rows, with the isoform 1 being a crucial element for the morphological maintenance of adult hair bundles and their function. The authors concluded that myosin 15 can be differentially recruited at different stereocilia and play a crucial role in initial growth and maintenance of the mechano-sensory transducer apparatus. This is an excellent manuscript tackling an important research question. It is also well presented and the numerous experiments are well executed.

I have some comments that should be addressed in order to clarify some of the findings/statements and improve the message of the manuscript (not listed in order of importance):

In the subsection “*Myo15*^*∆N/∆N*^ hair cells develop normal stereocilia bundles”. I am assuming that the SEM images are taken from young mice. If so, I am not convinced that the hair bundles in mutant mice develop normally. In Figure 1, the tip of the stereocilia of mutant hair cells is abnormally elongated (both of those that are visible) compared to that of controls. This abnormality is only highlighted by the authors later on in the manuscript, when describing adult/mature hair bundle. Have you made some quantification supporting your conclusion in young mice?

In the subsection “Isoform 1 localizes to the tips of shorter row mechanotransducing stereocilia”. Although very interesting, the statement that the localization of myosin 15 is linked to “sites under mechanical tension” is too speculative. I guess it would be quite difficult to prove this, but I guess you could look at mutants that lack the normal hair bundle resting tension or the tectorial membrane.

In the subsection “Isoform 2 is sufficient to traffic Whirlin and Eps8 within the hair bundle”. Consider that Eps8 is also present in the shorter stereocilia.

In the subsection “Isoform 1 influences the deflection sensitivity of MET machinery in IHCs“. Add the number of cells tested for Figure 5. Also, use “300 nm” instead “200 nm” to match the Figure (the Figure shows significant difference also at 300nm). Moreover, I cannot see any direct evidence supporting the statement that the mutation causes a change in the overall bundle stiffness. This statement should be validated by measuring bundle stiffness in both controls and littermate ko hair cells and values given in the test.

In the subsection “Isoform 1 is required for postnatal maintenance of shorter row mechanotransducing stereocilia”. “isoform 1 is not required for initial tip density formation...” This statement needs some additional verification – see above (subsection “*Myo15*^*∆N/∆N*^ hair cells develop normal stereocilia bundles”).

In the sixth paragraph of the Discussion. “We argue that membrane tensioning..., and that the absence of isoform 1 cripples this mechanisms”. I am not sure about this explanation; although possible it is unlikely for several reasons. While isoform 1 is mainly expressed during mature stages, the characteristic staircase structure of the bundle is defined during early postnatal days. Hair bundles with no resting tension, (e.g. Myo7a mutant mice) are able to initially develop a staircase structure.

Reviewer #2:

The study by Fang et al., presents the characterization of the deletion of a long isoform of Myo15, which is an uncoventional myosin required for function of hair cells. The results suggest that this isoform resides in different rows of stereocilia in inner hair cells in contrast to the shorter isoform of Myo15, and each Myo15 isoform plays a different role in assembly and maintenance of the structure of hair bundles. Overall, the study presents interesting findings and provides some insight into the function of the longer isoform of Myo15, which has been a bit of an enigma with regard to the function of the N-terminus present only in the longer isoform.

The manuscript is well written and I only have some minor concerns/comments:

The data in the first figure show that hair bundles in mice lacking the long isoform of Myo15 are normal, yet the mice are deaf. In a way, this figure is misleading because the SEM image is of an immature hair bundle at P4 and the tests for hearing function are performed much later (P14 and onwards). The reader is not shown the degeneration of hair bundles and likely cause for the hearing deficits until the penultimate figure of the study (Figure 6).

Figure 2 establishes the late onset of expression of the long isoform of Myo15, which corroborates nicely with the rest of the study. It is also satisfying to see that expression levels of each isoform do not change in the mouse mutants. So no major concerns with this figure, other than it is difficult to discern which isoform is more abundant. Presumably the relative expression level is relative to the level seen at P0, but not to each other.

The authors show that the long isoform is mainly localized to the shorter rows of inner hair cells. This pattern is not the case in outer hair cells. Is the localization pattern specialized only in cochlear inner hair cells? What about vestibular hair cells? Is this a universal feature of hair cells or something specific to inner hair cells? If the long isoform plays a special role in mechanotransduction, then it is hard to explain why it is found in the tallest row of stereocilia in outer hair cells. It is true that OHCs are not the main “transducers” in the cochlea, but rather help to amplify cochlear signals, however, this type of hair cell is, after all, the most often used cell type to measure transduction in mouse models. The authors mention the possibility of tension being created by the attachment of the tectorial membrane, so one might guess that this isoform will not be present in the tallest stereocilia of vestibular hair cells. Showing the localization in vestibular hair cells would make this idea, i.e. the long isoform is present at tips that experience tension, more compelling.

Figure 4 is key in showing a different pattern of localization for isoform 2. There are, however, a number of issues with this figure. All of the stages shown are from time points when isoform 2 is expressed at 10 fold lower levels as demonstrated in Figure 2. What about P0 or earlier, when isoform 2 is supposed to be exerting its function and important for development? In panel 4D, where are the shorter rows of stereocilia in this image? In panel 4F, whirlin is present in the shorter rows of stereocilia, yet the whirlin (and epsin8) signal is absent in the shorter rows shown in panels H and J. Why do these panels differ so much if they are of the same genotype?

In panel 4G: Unlike panel B, there appears to be whirlin protein in the shorter stereocilia. It appears that whirlin is localized to some extent independently of the shorter isoform of Myo15.

The changes in the MET currents are subtle and the authors emphasize the earlier opening of channels in the long isoform mutant, yet ignore the earlier onset of saturation of the current. The authors should also comment on this part of the phenotype and explain how it fits in with the idea of a missing spring element.

Also, the lower two rows of the OHCs have the long isoform present, yet the changes differ considerably from the IHCs (less current, slower adaptation). Why see these differences between the two hair cell types? The effects are challenging to interpret.

The final figure focuses on the ultrastructural changes in stereocilia upon deletion of the long isoform. It would behoove the authors to substitute a better image of the wild type tip link area that doesn't have a big, artifactual bleb. An outside reader may be misled into thinking that this membrane protrusion is normal. The stretching of the tips is certainly interesting, but there is no quantification of this defect. How often was it seen? In the Discussion, the authors say that “eruptions” are happening at the tips, but there is just one static TEM image in this study and there is no evidence that actin assembly is being dysregulated within this zone.

In general, there is a lot of speculation in the Discussion. Based on the evidence of this study, it is difficult to say whether Myo15 “regulates” actin filaments. The “eruptions” described above could be the simple extension of actin filaments into space during development. Evidence is lacking in terms of the time scale of this particular phenotype (and also the frequency of stretched tips is not shown). The actual thinning of stereocilia may be an indirect effect of the chronic loss of Myo15. Such a phenotype is also seen in other mouse mutants, such as the pirouette/GRXCR1 mutant.

Reviewer #3:

The manuscript by Fang et al. employ two new tools to study Myosin 15 function in hair cell development and physiology. Together these reveal the functional significance of alternative splicing of the Myosin 15 gene. The tools include antibodies that detect protein sequences encoding by Myosin15 exon two that are only found in Isoform1, and an engineered mouse model that mimics a human deafness allele that specific disrupts expression of Myosin 15 Isoform 1. Together these reagents reveal the interesting and significant findings of the paper; (1) that distinct protein Myosin15 isoforms contribute to hair cell development versus hair cell function and stereocilia homeostasis and (2) that distinct Myosin15 isoforms are trafficked to different rows of stereocilia. The figures are presented in a clear and concise manner, the data appears to be interpreted accurately and supports the conclusions drawn by the authors. Overall it is a good manuscript and is ready for publication.

The principal concern is primarily semantic. The title, “An unique N-terminal domain...” implies that this is the first report of myosin 15 alternative splice isoforms and the discovery of sequences unique to Isoform 1. However as stated in the Introduction, alternative splicing of the Mysoin15 gene was initially reported by [30], and humun mutations in exon2 that only affect Isoform 1 were previously identified (3, 10, 20, 37). A new title that accurately reflects the state of the field is recommended.

---

## [Author Response]

Reviewer #1:

[…] I have some comments that should be addressed in order to clarify some of the findings/statements and improve the message of the manuscript (not listed in order of importance):

*In the subsection “*Myo15^∆N/∆N^
*hair cells develop normal stereocilia bundles”. I am assuming that the SEM images are taken from young mice. If so, I am not convinced that the hair bundles in mutant mice develop normally. In*
Figure 1*, the tip of the stereocilia of mutant hair cells is abnormally elongated (both of those that are visible) compared to that of controls. This abnormality is only highlighted by the authors later on in the manuscript, when describing adult/mature hair bundle. Have you made some quantification supporting your conclusion in young mice?*

We are very grateful to the reviewer for spotting this inconsistency. Figure 1 as originally submitted showed a hair cell in which most of the second row stereocilia tips exhibited a more “tented” shape compared to the control.

Variations of IHC stereocilia tip shapes throughout the analyzed region of the cochlea (middle of the apical turn) have now been systematically quantified in a new Figure 7—figure supplement 3. In this analysis, we fit ellipses to the 2^nd^ row stereocilia tips using ImageJ and calculated aspect ratios (elliptical major axis divided by minor axis). We also quantified the staircase architecture of the hair bundle using measurements of row-to-row steps, a technique previously used to study *Tmhs* mutant mice ([57], PMID: 23217710). This data is presented in a new Figure 7—figure supplement 2. Full details of these quantification techniques have been added to the Materials and Methods.

Both of these analyses reveal that at P4, the stereocilia bundles of *Myo15^ΔN/ΔN^* IHCs are indistinguishable with controls. A morphological phenotype only becomes detectable later in development, by P8 (Figure 7—figure supplement 2 and Figure 7—figure supplement 3), consistent with the onset of isoform 1 expression around this age. Based upon these new data, we have replaced the image of a *Myo15^ΔN/ΔN^* IHC in Figure 2 with one more representative of the actual measured distribution. The age of the cells (P4) and their location along the cochlea (at the middle of the apical turn) are now referenced in the figure and legend.

It is worth stressing that the normal variations of stereocilia tip morphology along the analyzed region of the cochlea in wild-type IHCs are fundamentally different from the over-extended stereocilia observed in *Myo15^ΔN/ΔN^* IHCs (Figure 7). At P8, the second row stereocilia in *Myo15^ΔN/ΔN^* inner hair cells exhibit large variations of the lengths even within the same row (Figure 7), which are not observed in the control stereocilia bundles. The extent of non-uniformity of the mutant stereocilia is particularly well highlighted by this new analysis. We thank the Reviewer for suggesting that we make these measurements. Overall, they have strengthened our assertion that *Myo15^ΔN/ΔN^* hair bundles initially develop with a normal architecture.

In the subsection “Isoform 1 localizes to the tips of shorter row mechanotransducing stereocilia”. Although very interesting, the statement that the localization of myosin 15 is linked to “sites under mechanical tension” is too speculative. I guess it would be quite difficult to prove this, but I guess you could look at mutants that lack the normal hair bundle resting tension or the tectorial membrane.

We agree that our idea of isoform 1 associating with sites under mechanical tension remains entirely speculative at this present time. We wanted to highlight this association only as a possible hypothesis for why isoform 1 localizes differently to the tallest stereocilia in IHCs vs OHCs. This idea has been moved from the Results to the very end of the Discussion, as we feel that it is still worth proposing. The experiments suggested by the Reviewer are an interesting approach to exploring this relationship and are an area for future investigation.

In the subsection “Isoform 2 is sufficient to traffic Whirlin and Eps8 within the hair bundle”. Consider that Eps8 is also present in the shorter stereocilia.

We assume that the reviewer is referring to the mechanism of how Eps8 is localized to shorter row stereocilia. The quantification in Figure 4 shows that there is some isoform 2 present on the shorter stereocilia rows at P7, but that isoform 1 is the major species at this location. The presence of smaller quantities of isoform 2 is consistent with our detection of whirlin and Eps8 on the 2^nd^ IHC stereocilia row at P7.

An updated Figure 4—figure supplement 1 has been provided to show Eps8 labeling on the shorter stereocilia in P7 *Myo15ΔN/ΔN* IHCs (Figure 4—figure supplement 1). In our hands, the labeling of shorter stereocilia with Eps8 immunofluorescence is very weak.

The text in the subsection “Isoform 2 is sufficient to traffic Whirlin and Eps8 within the hair bundle” has also been modified to highlight that the segregation of Eps8 and whirlin to the tallest stereocilia row is not 100% at P7.

*In the subsection “Isoform 1 influences the deflection sensitivity of MET machinery in IHCs“. Add the number of cells tested for*
Figure 5*. Also, use “300 nm” instead “200 nm” to match the Figure (the Figure shows significant difference also at 300nm). Moreover, I cannot see any direct evidence supporting the statement that the mutation causes a change in the overall bundle stiffness. This statement should be validated by measuring bundle stiffness in both controls and littermate ko hair cells and values given in the test.*

As requested, we have added the number of cells examined to the main text. The main text now reads as, “Maximal MET current amplitudes were not statistically different between mutant *Myo15^ΔN/ΔN^* (n=12) and control *Myo15*^*+/ΔN*^ IHCs (n=10) (Figure 5). However, the responses to small bundle deflections (150-300 nm) were significantly larger in mutant *Myo15^ΔN/ΔN^* IHCs, indicating an increased deflection sensitivity of the transduction apparatus in the absence of isoform 1 (Figure 5)”. We also added the number of OHCs tested, *Myo15^ΔN/ΔN^* (n=9), *Myo15*^*+/ΔN*^ (n=7), in the relevant sentence at the end of that paragraph.

We did not intend to suggest that the overall stiffness of the hair bundle is affected in *Myo15^ΔN/ΔN^* hair cells. Instead, we referred to the stiffness of the theoretically defined gating spring element attached to the MET channel. In young postnatal mammalian inner hair cells, even complete disruption of MET machinery with BAPTA does not result in significant changes to the overall bundle stiffness ([Bibr bib28a]). Measurements of the non-linear mechanical compliance of the MET machinery in mammalian auditory hair cells also varies in the hands of the same group (Kennedy et al., 2005, PMID:15696193; Beurg et al., 2008, PMID:18178649). These measurements are certainly beyond the scope of our current characterization of *Myo15^ΔN/ΔN^* mutants. To avoid confusion, we have rephrased the relevant text as follows:

“This sensitivity depends on the mechanical stiffness of a theoretical “gating spring” that is connected to the MET channel (25). Our data show that isoform 1 is not essential for MET responses in IHCs but may contribute to the stiffness of this gating spring”.

*In the subsection “Isoform 1 is required for postnatal maintenance of shorter row mechanotransducing stereocilia”. “isoform 1 is not required for initial tip density formation...” This statement needs some additional verification – see above (subsection “Myo15*^*∆N/∆N*^
*hair cells develop normal stereocilia bundles”).*

Our current manuscript shows that the tip-density is clearly present in both wild-type and isoform 1-null *Myo15^ΔN/ΔN^* hair cells at P7. We have now additionally examined the tip- density in *Myo15^ΔN/ΔN^* hair cells at P3, and found that the tip-density is also detected at this younger age (see Figure 8). Combined with our new analysis of stereocilia morphology, these data further strengthen our assertion that stereocilia bundles initially develop normally (at least to P4) in the absence of isoform 1.

Author response image 1.TEM images of ultrathin sections cut along the longitudinal stereocilia axis of *Myo15*^*+/ΔN*^ (A) and *Myo15**^ΔN/ΔN^* (B) inner hair cells at P3. Scale bars are 500 nm.**DOI:**
http://dx.doi.org/10.7554/eLife.08627.020

In the sixth paragraph of the Discussion. “We argue that membrane tensioning..., and that the absence of isoform 1 cripples this mechanisms”. I am not sure about this explanation; although possible it is unlikely for several reasons. While isoform 1 is mainly expressed during mature stages, the characteristic staircase structure of the bundle is defined during early postnatal days. Hair bundles with no resting tension, (e.g. Myo7a mutant mice) are able to initially develop a staircase structure.

Our hypothesis is that tension applied to the tips of mature mechanosensitive (2^nd^ row) stereocilia, via tip links, can regulate actin polymerization in the region of the tip-density. We agree that our current wording gives the wrong impression that tension might also regulate actin polymerization during stereocilia development. As the Reviewer points out, the existing evidence argues against this being the case.

We have reworded this sentence to emphasize that we are referring to mature stereocilia and have added citations to recent reviews that describe how plasma membrane tension regulates actin polymerization in other systems, and potentially also in stereocilia. It now reads: “We argue that membrane tension is instead a critical factor controlling actin polymerization at the tips of mature mechanosensitive stereocilia (2; 16) and that isoform 1 may contribute to a tension-sensing mechanism”.

Reviewer #2:

[…] The manuscript is well written and I only have some minor concerns/comments:

*The data in the first figure show that hair bundles in mice lacking the long isoform of Myo15 are normal, yet the mice are deaf. In a way, this figure is misleading because the SEM image is of an immature hair bundle at P4 and the tests for hearing function are performed much later (P14 and onwards). The reader is not shown the degeneration of hair bundles and likely cause for the hearing deficits until the penultimate figure of the study (*Figure 6*).*

To address the reviewer’s concern we have rearranged the presentation of data in Figures 1 and 2. The qPCR data has been moved to a new Figure 1 alongside the ABR data, whilst a new Figure 2 is now dedicated to the SEM data at P4. These changes have entailed some rearrangement of the main Results section to account for the presentation order. The overall content and conclusions drawn in these sections remains unchanged. The flow of the results text is much improved and we thank the Reviewer for this suggestion.

To specifically alert readers to the likely cause of deafness, the following sentence has been added to the Results: “Although hair bundles in IHCs and OHCs initially developed normally at P4, they subsequently degenerated (Figure 6) and this likely contributes to the deafness observed in *Myo15ΔN/ΔN* mice at P14 and older (Figure 1)”.

Figure 2
*establishes the late onset of expression of the long isoform of Myo15, which corroborates nicely with the rest of the study. It is also satisfying to see that expression levels of each isoform do not change in the mouse mutants. So no major concerns with this figure, other than it is difficult to discern which isoform is more abundant. Presumably the relative expression level is relative to the level seen at P0, but not to each other.*

The relative expression values quoted in Figure 2 (now Figure 1 in the revised manuscript) are referenced to a housekeeping gene (*Tbp*), which encodes the TATA-box binding protein, and then subsequently to P0 for each isoform (ΔΔC_T_ method). This does not allow for direct comparisons between the abundance of each isoform. We have reanalyzed our raw qPCR data to present the different transcripts referenced to *Tbp* only (ΔC_T_ method), and have added this data to Figure 1—figure supplement 1. These data show that isoform 2 mRNA is more abundant than isoform 1 at P0, but that isoform 1 becomes the most abundant transcript from P6 onwards.

The axes in Figure 2 have been relabeled to highlight that they derive from ΔΔC_T_, to distinguish them from the supplemental data, which are ΔC_T_.

The authors show that the long isoform is mainly localized to the shorter rows of inner hair cells. This pattern is not the case in outer hair cells. Is the localization pattern specialized only in cochlear inner hair cells? What about vestibular hair cells? Is this a universal feature of hair cells or something specific to inner hair cells? If the long isoform plays a special role in mechanotransduction, then it is hard to explain why it is found in the tallest row of stereocilia in outer hair cells. It is true that OHCs are not the main “transducers” in the cochlea, but rather help to amplify cochlear signals, however, this type of hair cell is, after all, the most often used cell type to measure transduction in mouse models. The authors mention the possibility of tension being created by the attachment of the tectorial membrane, so one might guess that this isoform will not be present in the tallest stereocilia of vestibular hair cells. Showing the localization in vestibular hair cells would make this idea, i.e. the long isoform is present at tips that experience tension, more compelling.

Why isoform 1 localizes to the tips of the tallest stereocilia row in OHCs, but not IHCs, remains puzzling. The “tension hypothesis” is the best explanation so far, since all of these sites with isoform 1 experience tension as the stereocilia are deflected. This idea is still speculative, as we stressed to Reviewer 1, however we do believe this hypothesis is worth presenting to generate discussion in the field and guide future experiments.

Following the reviewer’s suggestion, we have examined PB888 labeling of isoform 1 in wild-type mouse utricles at P14. We observe PB888 immunofluorescence labeling primarily in striolar hair cells where it is present at the tips of all stereocilia rows (see Figure 9). PB888 labeling of extra-striolar hair cells is weak and sporadic (data not shown). These data are entirely consistent with isoform 1 localizing to sites under tension, since the tips of the tallest stereocilia rows in vestibular hair cells are coupled to the kinocilium, which is ultimately connected to the otoconial matrix.

Author response image 2.PB888 labeling of isoform 1 in a striola hair cell from a P14 utricle. Phalloidin was used to stain the stereocilia actin core. PB888 is detected at the tips of all stereocilia rows. Scale bar is 5 µm.**DOI:**
http://dx.doi.org/10.7554/eLife.08627.021

We would like to clarify that we are not claiming any special role for myosin 15 isoform 1 in mechano-electrical transduction (MET). Quite the opposite: we only detect minor changes to MET currents in the absence of isoform 1 (Figure 5) and these could result from subtle changes to the protein complexes that presumably couple the membrane or MET channel complex to the stereocilia actin core. According to our data, the major function of isoform 1 is to maintain the actin cytoskeleton in mechanotransducing stereocilia (i.e. those with active MET). This function is conserved in both IHCs and OHCs, despite minor differences in isoform 1 localization in these cells.

Figure 4
*is key in showing a different pattern of localization for isoform 2. There are, however, a number of issues with this figure. All of the stages shown are from time points when isoform 2 is expressed at 10 fold lower levels as demonstrated in*
Figure 2*. What about P0 or earlier, when isoform 2 is supposed to be exerting its function and important for development?*

Images for isoform 1 (PB888) and isoform 2 (PB48 in the isoform 1-null background) at P1 are available in Figure 3—figure supplement 1 and Figure 4—figure supplement 1. At this earlier age we do detect isoform 2 on the tallest stereocilia row, consistent with our hypothesis that it is driving stereocilia elongation.

In panel 4D, where are the shorter rows of stereocilia in this image?

This image was taken from “behind” the hair cell, with the tallest row of stereocilia nearest to the microscope objective. To further clarify that PB48 labeling is absent from the shorter rows, please see the additional images below (Figure 10). We have added a cropped portion of this data (panel B, lower right) to panel 4D in the main manuscript.

Author response image 3.PB48 labeling (green) of myosin 15 in normal hearing (A) and isoform 1-null (B) IHCs at P14*.* In the isoform 1-null background, PB48 labeling is primarily restricted to the 1^st^ (tallest) stereocilia row (B), compared to the normal-hearing control where both 1^st^ and 2^nd^ / 3^rd^ rows are labeled (A). Rhodamine phalloidin is overlaid in red. Scale bar, 5 µm.**DOI:**
http://dx.doi.org/10.7554/eLife.08627.022

In panel 4F, whirlin is present in the shorter rows of stereocilia, yet the whirlin (and eps8) signal is absent in the shorter rows shown in panels H and J. Why do these panels differ so much if they are of the same genotype?

We agree that the current figure layout is confusing. To improve figure clarity, we have moved the ages and genotypes to within each image, and furthermore grouped images that are logically connected into combined panels.

The images refered to are all from normal-hearing *Myo15^+/ΔN^* cochleae, but Panel F is P7 (whirlin, now panel D), whilst panels H (whirlin, now E) and J (Eps8, now F) are from P14 cochleae. The Reviewer is correct that in normal mice, both Eps8 and whirlin (Figure 4 and Figure 4—figure supplement 1) are detected on the 2^nd^ row IHC stereocilia at P7, and that this labeling is not evident in older animals at P14 (Figure 4, top row). These age-dependent changes in the localization of whirlin and Eps8 are identical to previously published data in wild-type mice (5; 12; 34; 23). The changes in Eps8/whirlin also mirror the time course of isoform 2 loss from the shorter stereocilia rows between P7 and P14 (compare Figure 4, lower rows).

In panel 4G: Unlike panel B, there appears to be whirlin protein in the shorter stereocilia. It appears that whirlin is localized to some extent independently of the shorter isoform of Myo15.

The Reviewer is absolutely correct that a weak whirlin signal is present on the shorter stereocilia of isoform 1-null hair cells at P7. This is entirely consistent with the small amounts of myosin 15 isoform 2 that we detect at P7 (quantified in Figure 4) at this location.

The existing manuscript does allude to this in the subsection “Myosin 15 isoforms are differentially trafficked within the hair bundle”: “This indicates that the majority of myosin 15 in the second row is isoform 1 and that isoform 2 is the minor species at this location”.

The changes in the MET currents are subtle and the authors emphasize the earlier opening of channels in the long isoform mutant, yet ignore the earlier onset of saturation of the current. The authors should also comment on this part of the phenotype and explain how it fits in with the idea of a missing spring element.

Following the reviewer’s suggestion, we have added the following sentence to subsection “Isoform 1 influences the deflection sensitivity of MET machinery in IHCs”, “A stiffer gating spring would transmit the maximal opening force to the MET channel at a smaller bundle deflection, resulting in earlier saturation of the current-displacement relationship. This was indeed observed in *Myo15^ΔN/ΔN^* IHCs (Figure 5)”.

Also, the lower two rows of the OHCs have the long isoform present, yet the changes differ considerably from the IHCs (less current, slower adaptation). Why see these differences between the two hair cell types? The effects are challenging to interpret.

We agree that the different effects of isoform 1 deficiency upon MET currents in mutant IHCs and OHCs are hard to interpret; especially so if one assumes that the mechanotransduction mechanisms are identical in all vertebrate hair cells. There is however, growing evidence of essential differences between IHCs and OHCs in the MET apparatus itself. A notable example is the tonotopic distribution of the single MET channel conductance in OHCs, but not in IHCs (Beurg et al., 2006, PMID: 17065441).

Furthermore, the shaker 2 mutation, which we show in the current manuscript affects both myosin 15 isoforms, causes different effects on the hair bundle staircase and MET currents in IHCs versus OHCs ([51], PMID: 19339598). Without the exact knowledge of all proteins that populate the different stereocilia tips of OHCs and IHCs, it is premature to speculate on the detailed mechanisms of MET changes following isoform 1 loss in these different sensory cell types.

We would like to clarify that fast adaptation in *Myo15^ΔN/ΔN^* OHCs is not slower than control OHCs. We state in the subsection “Isoform 1 influences the deflection sensitivity of MET machinery in IHCs”, “The time constant of adaptation was not affected in either IHCs or OHCs of *Myo15^ΔN/ΔN^* mice compared to *Myo15*^*+/ΔN*^ controls”. This is also illustrated in Figure 5. In fact, the extent of this adaptation (the percentage of the adapted current) is even larger in mutant OHCs (Figure 5), indicating that the adaptation machinery is likely to be unaffected by isoform 1 deficiency.

The final figure focuses on the ultrastructural changes in stereocilia upon deletion of the long isoform. It would behoove the authors to substitute a better image of the wild type tip link area that doesn't have a big, artifactual bleb. An outside reader may be misled into thinking that this membrane protrusion is normal.

We have replaced the image in Figure 7, as requested.

The stretching of the tips is certainly interesting, but there is no quantification of this defect. How often was it seen? In the Discussion, the authors say that “eruptions” are happening at the tips, but there is just one static TEM image in this study and there is no evidence that actin assembly is being dysregulated within this zone.

We have now quantified the shape factor (aspect ratio of an ellipse fitted to the stereocilia tip) of 2^nd^ row stereocilia in both normal hearing *Myo15^+/ΔN^* and mutant *Myo15^ΔN/ΔN^* IHCs at P4, P6 and P8 (see the new Figure 7—figure supplement 3), and thank the Reviewer for requesting this important addition to our manuscript.

At P4, we do not detect a difference in the distribution of aspect ratios. However by P6 and P8, there are statistically significant increases in the tip aspect ratios of *Myo15^ΔN/ΔN^* stereocilia. Critically, we observed stereocilia at P8 with aspect ratios >> 2 (note the compressed axis in Figure 7—figure supplement 3), and these correspond to the unusually over-elongated stereocilia tips shown in Figure 7.

It is possible that the tips are stretched, but we favor the hypothesis that abnormalities in the actin cytoskeleton are the driving force behind the membrane distension. TEM images of sections through over-elongated tips (Figure 7) reveal actin filaments in close apposition beneath the membrane, which is consistent with the actin cytoskeleton providing force to distend the membrane, similar to an extending lamellipodium of a motile cell. Since this type of more extreme protrusion morphology was not observed at P4 or P6 (Figure 7—figure supplement 3), the actin cytoskeleton must have remodeled to power this morphological change.

Our SEM measurements provide a static snapshot of what is a likely dynamic process. Future experiments will use live cell imaging experiments to investigate this phenomenon further, but these are beyond the scope of our current study.

We have removed the word “eruption” from the revised manuscript.

In general, there is a lot of speculation in the Discussion. Based on the evidence of this study, it is difficult to say whether Myo15 “regulates” actin filaments. The “eruptions” described above could be the simple extension of actin filaments into space during development. Evidence is lacking in terms of the time scale of this particular phenotype (and also the frequency of stretched tips is not shown).

We are grateful to the Reviewer for encouraging us to quantify the morphological phenotype of mutant stereocilia. These new data (Figure 7—figure supplement 2 and Figure 7—figure supplement 3) allow us to address these points directly.

The over-elongation of stereocilia involves the extension of actin filaments (Figure 7), however the new analysis of stereocilia morphology does not support a model where actin filaments simply overextend during development. The overall staircase architecture of mutant hair cells is indistinguishable from normal samples at P4 (Figure 7—figure supplement 2) and over- elongated tips were not observed in either mutant or wild-type at this age (Figure 7—figure supplement 3).

This indicates that at P4, the stereocilia actin cytoskeleton is still similarly regulated in mutants compared with wild-type controls. However by P8, over-elongated stereocilia tips are detected in mutant hair cells but are absent from littermate controls (Figure 7—figure supplement 3). This demonstrates that the changes observed in mutant hair cells are not a recapitulation of the normal developmental program. We note that the onset of stereocilia tip over-elongation in mutant hair cells occurs within a similar time window to when isoform 1 normally localizes to the stereocilia bundle.

We have taken this opportunity to revise parts of the Discussion to try and clarify the argument for myosin 15 regulating the actin cytoskeleton and how this might operate in hair cells. Clearly, we do not have a detailed biochemical mechanism, so we use “regulate” in the broadest possible sense to include either direct, or indirect regulation of actin filaments by myosin 15.

As a comparison, the Rho family of GTPases were originally identified as molecules that could regulate the actin cytoskeleton and generate stress fibers, lamellipodia and filopodia (Nobes and Hall, 1995, PMID: 7536630). It was not until later that the multiple effector proteins and pathways that mediate this effect were identified.

The actual thinning of stereocilia may be an indirect effect of the chronic loss of Myo15. Such a phenotype is also seen in other mouse mutants, such as the pirouette/GRXCR1 mutant.

The mechanism of stereocilia thinning is unknown, except to say that it involves the loss/depolymerization of actin filaments from within the core. It is unlikely due to a chronic loss of myosin 15, since *Myo15^sh2/sh2^* hair cells (which lack both isoforms in the stereocilia bundle) have nearly identical stereocilia thicknesses in all rows within the hair bundle ([51], PMID:19339598). Furthermore, unlike the *pirouette* mouse that exhibits thinning of all stereocilia rows, we observe a specific thinning of the mechanosensitive shorter rows only. The *Myo15^ΔN/ΔN^* phenotype is therefore unlikely due to general hair cell degeneration, since the tallest stereocilia row appears unaffected (Figure 6).

Reviewer #3:

*[…] The principal concern is primarily semantic. The title, “An unique N-terminal domain...” implies that this is the first report of myosin 15 alternative splice isoforms and the discovery of sequences unique to Isoform 1. However as stated in the Introduction, alternative splicing of the Mysoin15 gene was initially reported by*
[30]*, and humun mutations in exon2 that only affect Isoform 1 were previously identified (*[3]*,*
[10]*,*
[20]*,*
[37]*). A new title that accurately reflects the state of the field is recommended.*

The title has been revised to “The 133-kDa N-terminal domain enables myosin 15 to maintain mechanotransducing stereocilia and is essential for hearing”.

Mutations in exon 2, which encodes the N-terminal domain unique to isoform 1, have previously been identified in several human pedigrees segregating recessive hereditary deafness, *DFNB3*. Our manuscript contains the first experimental demonstration that a mutation in exon 2 causes deafness. In addition, we provide the first mechanistic insight into the resulting hair cell pathology.

Other significant changes:

We realized that the targeting vector design presented in Figure 1—figure supplement 1 was difficult to interpret, and did not adequately explain how the Southern blot screening strategy worked. We revised the figure to emphasize the introduction of a novel *Asp718* restriction endonuclease site that was used for screening ES clones for homologous recombination, and we hope the clarity is improved. The legend to Figure 1—figure supplement 1 and the Materials and methods has been modified accordingly.